# EXPLEME: A STUDY IN MEME INTERPRETABILITY, DIVING BEYOND INPUT ATTRIBUTION

## ABSTRACT

Memes, originally created for humor and social commentary, have evolved into vehicles for offensive and harmful content online. Detecting such content is crucial for upholding the integrity of digital spaces. However, classification of memes as offensive or other categories often falls short in practical applications. Ensuring the reliability of these classifiers and addressing inadvertent biases during training are essential tasks. While numerous input-attribution based interpretability methods exist to shed light on the model's decision-making process, they frequently yield insufficient and semantically irrelevant keywords extracted from input memes. In response, we propose a novel, theoretically grounded approach that extracts meaningful "tokens" from a predefined vocabulary space, yielding both relevant and exhaustive set of interpretable keywords. This method provides valuable insights into the model's behavior and uncovers hidden meanings within memes, significantly enhancing transparency and fostering user trust. Through comprehensive quantitative and qualitative evaluations, we demonstrate the superior effectiveness of our approach compared to conventional baselines. Our research contributes to a deeper understanding of meme content analysis and the development of more robust and interpretable multimodal systems.

## 1 INTRODUCTION

In recent times, memes have emerged as a ubiquitous form of online expression, blending humor, satire, and social commentary to encapsulate complex ideas in a single image or short video. While originally created to disseminate humor, it is often misused to perpetuate societal harm(Kiela et al., 2021). A significant portion of the meme ecosystem is tainted with content that is offensive, hateful, or even dangerous. Therefore, it is crucial to develop effective tools for the automated detection of offensive memes, to preserve the integrity of online spaces.

However, a simple classification of memes as offensive is often insufficient. Making the system interpretable is paramount as it can elucidate whether the automatic detection system learns from spurious correlations in the data or whether it can reliably classify a meme as offensive. This clarity aids in enhancing user trust through transparency. Further interpretability methods help users to know if the model acquired some kind of inadvertent biases while training.

Existing input-attribution based explainability methods like LIME(Ribeiro et al., 2016), SHAP(Guo et al., 2019), and GradCAM(Selvaraju et al., 2019) work pretty well in practice but suffer from two issues, *viz.* i) *Semantic Irrelevance:* The input keywords that are attributed to model predictions are often semantically irrelevant to the input meme, making it hard for humans to assess their effect on model's behavior; and ii) *Lack of diversity:* Existing methods work on the input space, which fundamentally lacks many important words that would have explained the model and its prediction much better.

In Figure 1, we show our method EXPLEME and compare it with Integrated Gradient. The tokens retrieved by input attribution are 'Polish', 'Chemical', and 'Shit', which are irrelevant to the hidden meaning of the meme which is associated with 'Antisemitism'. In contrast, our method can reliably consult a large set of vocabulary space and through a four-step process, retrieves much relevant and exhaustive set of keywords (e.g. 'Holocaust', 'Auschwitz' etc.). This comprehensively shows the superiority of our proposed methods over conventional baselines.

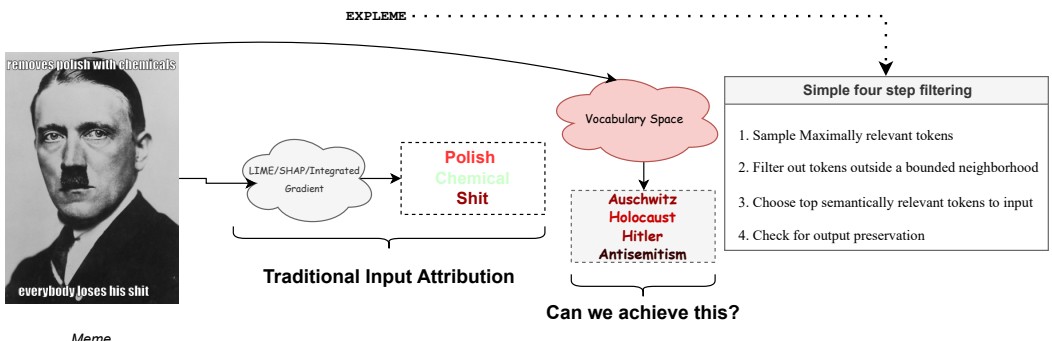

Figure 1: We posit our method: EXPLEME, as a simple four-step filtering mechanism to retrieve tokens related to an input meme from a global vocabulary space as opposed to traditional input attribution. Observe the richer quality of outputs that can be obtained by EXPLEME.

We enumerate the major contributions/attributes of our current work as follows:

1. We propose a theoretically grounded technique[1] that could explain a model's behavior by retrieving 'tokens' from a global vocabulary space. The retrieved tokens are compared with input attribution based baselines and found to carry both 'faithful' representation of the input meme as well as semantically relevant information.

2. Our method is extensively evaluated with respect to both automatic and human evaluation. A detailed analysis is performed to assess its effective nature.

3. Though we show our method as applied in the domain of internet memes, it is in principle, model and task-agnostic[2].

## 2 RELATED WORK

**Multimodal offensiveness detection.** In the realm of Natural Language Processing (NLP), previous research has primarily concentrated on identifying offensive content (Waseem & Hovy, 2016; Sarkar et al., 2021), addressing cyberbullying (Van Hee et al., 2018), hate speech(Caselli et al., 2021), and similar issues within social media posts (Roberts et al., 2012). Nevertheless, these computational methods have predominantly been evaluated using textual data. Turning to visual offensiveness detection, earlier investigations have centered on identifying sexually explicit images (Duan et al., 2001; Ganguly et al., 2017). The pivotal moment came when Kiela et al. (2021) introduced a set of benchmarks and released the Facebook Hateful meme dataset, which ignited research in this field. This led to a number of research on detecting offensiveness in multimodal media (Yang et al., 2022), particularly in memes (Sharma et al., 2020). Suryawanshi et al. (2020) used an early fusion method for combining visual and textual features, leading to more accurate detection. Chen & Pan (2022) stacked visual features, object tags, and text features to Vision-Language Pre-Training Model with anchor points to detect offensive memes. While these models are proven to be useful for predictions, their outputs are not interpretable and cannot be reliably used in real-world use cases.

**Multimodal Interpretability.** Recently, there have been a notable number of multimodal models (Ding et al., 2021; Du et al., 2022; Li et al., 2023; Liu et al., 2023b; Zhu et al., 2023) for various tasks. However, there is a dearth of research on generating explanations or justifications around their predictions. Researchers predominantly relied on interpretability techniques. LIME (Ribeiro et al., 2016) explains predictions of any classifier by fitting a sparse linear model locally around the instance being explained. It converts the instance into a binary vector, indicating the presence/absence of interpretable components (like words). SHAP (Lundberg & Lee, 2017a) explains machine learning model predictions by computing Shapley values from game theory. These values represent each

---

[1]Codes are available at:https://anonymous.4open.science/r/Expleme-E8BE/

[2]It is out of the scope of this paper to show its application to other tasks

feature's contribution to a prediction. SHAP unifies several local explanation methods like LIME into a single framework with theoretical properties. Gradient heatmap (Guo et al., 2019) explains predictions by computing gradients of the model output with respect to the input features. However, recent years have witnessed a shift in the focus of interpretability research, recognizing the potential for generating natural language explanations for both unimodal and multimodal systems (Kayser et al., 2021). Instead of traditional end-to-end training, Koh et al. (2020) first predicted concepts and used those to predict the labels such that the model can be interpreted by changing the concepts. There exist some natural-language-based techniques like wt5 (Narang et al., 2020). However, wt5 is available for text-only systems. NLX-GPT (Sammani et al., 2022) bridges the gap between text-based and multimodal natural language generation. Some recent methods generate explanations for multimodal systems. A cross-modal attention, which attends to the distinguishing features between text and image modalities, is used in the transformer encoder for sarcasm explanation (Desai et al., 2021). Sharma et al. (2022) generates explanations for visual semantic role labeling in memes. These methods can generate explanations for the behavior of multimodal models. However, the existing methods cannot fully explain the model behavior that is not directly related to the input but has some implicit meaning. Therefore, we attempt to address the issues by sampling keywords that are related to the input and faithful to the inner workings of the model.

## 3 METHODOLOGY

The proposed systems combine a multimodal encoder for an input meme and a language model (LM). The incorporation of LM enables us to retrieve the set of explainable out-context keywords that are helpful in interpreting the system and its outcome.

### 3.1 SYSTEM DESIGN

Our system follows a two-step strategy, i.e. i) Multimodal encoding followed by ii) Classifying via a language model (LM). We elaborate the steps in details:

**Multimodal encoding.** Let $M$ denote the input meme, consisting of an image $V$ and accompanying text $T$. We utilize a pre-trained and frozen CLIP model (Radford et al., 2021) to obtain textual ($f_t$) and visual ($i_t$) representations. These features, with dimensions $\mathbb{R}^{m \times 1}$ and $\mathbb{R}^{n \times 1}$ respectively, are used to generate a multimodal representation $M_t \in \mathbb{R}^{o \times 1}$ (here, $m = n = 512$).

The fusion process employs trainable weight matrices $U$ and $V$ with dimensions $\mathbb{R}^{m \times ko}$. The multimodal representation is calculated as follows: $M_t = AveragePool(U^T f_t \circ V^T i_t, k)$, where $\circ$ denotes element-wise multiplication, and $k$ represents the stride for the overlapped window used in the pooling operation. This encoding scheme, inspired by a similar approach (Bandyopadhyay et al., 2023), maintains high performance with a low parameter count.

**Using LM as the classifier cum verbalizer.** We utilize a GPT2 model(Radford et al., 2019) as the classifier. The multimodal representation $M_t$ is transformed via a Feed Forward Neural Network (FFN) into $m \in \mathbb{R}^{1 \times 1024}$, a dimension similar to the GPT2 token embedding. Another FFN projects $M_t$ onto the corresponding label space, resulting in $l \in 0, 1$.

$$l_i = argmax(FFN(M_t), dim = 1)$$

To address gradient information loss, Gumbel Logits ($gl$) are introduced:

$$gl = F.gumbel\_softmax(logits, tau = 1, hard = True)$$

A dictionary ($dix$) is established to convert labels into keywords: $dix = 0 : normal, 1 : offensive$. The final label with preserved gradient information is obtained as:

$$lab = SumPool(gl \circ E[dix[l]]),$$

where $E$ represents the GPT2 token-embedding function.

The GPT2 model takes inputs including knowledge texts ($kt_i$) separated by **[KB]** tokens and the meme $caption$ (automatically generated via OFA (Wang et al., 2022) module) separated by

*[CAPTION]'* token. These texts are converted into token embeddings:

$$f\_1 = E([CAPTION]caption[KB]kt\_1[KB]kt\_2)$$

Finally, concatenating $f_1$ with $M_t$ and $lab$, the input is fed into GPT2. The model reconstructs the next tokens and predicts the final label of the meme in natural language as the last token in the output. The system architecture is visualized in the Appendix Section C with a more detailed explanation.

## 3.2 RETRIEVING KEYWORDS BEYOND INPUT SPACE

Obtaining a comprehensive collection of human-comprehensible keywords that effectively encapsulate the operational characteristics of a classifier when presented with a given input typically poses a challenge. One plausible approach to address this issue involves the utilization of input-attribution-based methodologies such as Integrated Gradient. Such techniques serve to highlight specific parts of the input data that bear significance in the classifier's decision-making process. However, it is noteworthy that these methodologies do not yield an extra set of tokens which was not in the input space. This can be a limiting factor in describing the behavior of the model, as it is illustrated by an example in Figure 1.

The process of extracting the set of relevant keywords involves four filtering steps starting from the vocabulary set of the language model (LM).

1. **Maximally Relevant:** First, we filter out the keywords that are given very low probability by the LM. Given the concatenation of $f_1$, $lab$, and $m$ as input to the GPT2, we extract TopK tokens (denoted as set $T$) from the GPT2 predictions. $T = argmax_k P(t_i|f_1, lab, m)$, where $P(.)$ refers to the probability distribution over vocabulary $V$.

2. **Alignment vs Optimization:** The set of extracted keywords from the first step (*their embedding is denoted as $e$*) should be such that they belong in a special $\epsilon$ neighborhood. This is an additional filter that theoretically ensures the set of keywords does not possess redundant information while also not completely alienated from the optimizing model prediction. The definition and interpretation of this is presented in Section 3.3.

3. **Semantically Relevant:** In this step, we filter additional keywords that are semantically irrelevant to the input meme. Practically, the set of keywords extracted from the second step is passed through the CLIP Text encoder. Similarly, the meme is encoded using the CLIP Vision encoder. We take the dot product of the vector output from these two encoders and only select the top 20 tokens out of them. CLIP being trained with contrastive learning only preserves tokens that are semantically relevant to the input meme.

4. **Prediction Preserving :** The fourth step is the most stringent one. First, we use the trained LM in inference mode to generate knowledge text by passing extracted tokens as knowledge tokens. Next, together with the extracted tokens, we pass their generated knowledge text to the LM. If the model predicts the same class as it predicted before, we call the passed token *prediction preserving*. If the passed tokens flip the actual prediction then we can confidently say that the token does not have enough importance for the model's behavior and thus it cannot be adequately faithful to the model's behavior. We filter out only four keywords after this step by sorting with respect to the log-likelihood of the predicted tokens in decreasing order.

## 3.3 ALIGNMENT VS OPTIMIZATION TRADEOFF

In the pursuit of optimizing machine learning models, we often encounter the challenge of striking the right balance between the alignment of information vectors and optimization efficiency. To explore this delicate tradeoff, we introduce the following theorem.

**Theorem.** Assuming that our objective function $f(m) = \hat{y}$ is strongly convex, and considering non-zero real column vectors $e$ (also non-negative) and $m$, where $m$ represents the multimodal embedding of the input meme, and $\nabla_m \hat{y}$ is the gradient of the final network prediction with respect to $m$, our theorem states that, with very small step size, the condition $e^T \cdot \nabla_m f(m^+) > e^T \cdot \nabla_m f(m)\rho$, where $\rho > 1$, holds true.

This theorem carries substantial empirical implications:

i) If we sample $e$ such that $e^T \cdot \nabla_m f(m) > 0$, implying alignment between $e$ and $\nabla_m f(m)$, moving $m$ in the direction of $e$ aids optimization. As demonstrated by the left-hand side (LHS) of the inequality, successive gradient ascents on $m$ progressively reduce the angle between $e$ and $\nabla_m f(m)$ until they become aligned. This observation underscores the utility of $e$ throughout the optimization process.

ii) With $\nabla_m f(m)$ being smooth and differentiable, when $e^T \cdot \nabla_m f(m) \to 0$, we find that $e^T \cdot \nabla_m f(m^+) > 0$. Even as $e$ and $\nabla_m f(m)$ approach near-orthogonality, indicative of $e$ carrying diverse information rather than perfect alignment with the gradient, the positive value of $e^T \cdot \nabla_m f(m^+)$ signifies $e$ as beneficial for subsequent gradient-based optimization steps w.r.t $m$. We term this phenomenon the *'Alignment vs. Optimization Tradeoff Criteria'* In practical applications, this serves as a filtering mechanism to retain tokens relevant to regions where $e^T \cdot \nabla_m f(m) \to 0^3$. This theoretically grounded motivation significantly enhances our ability to extract pertinent and improved tokens, as shown through careful ablation experiments. This theorem and its implications shed light on the intricate relationship between vector alignment and optimization effectiveness. The proof of this theorem is shown in Appendix Section A

### 3.4 ALGORITHM

In this section, we illustrate the algorithm of our proposed four-step retrieval mechanisms1 to retrieve explainable keywords that are outside the input space.

---

**Algorithm 1** Algorithm: Retrieve explainable keywords outside input space

```
explain_out = [] ;                                              /* Final token placeholder */
first_stage = [] ;                       /* Placeholder or TopK & ε neighborhood constraint */
r_clip ← Meme Image Embedding from CLIP

  { t_i } ← Top-k tokens from Vocabulary set V ;                      /* TopK Filtering */
for t_i ∈ {t_i} do
    e_i ← GPT2Embedding( t_i )
    if ‖e_i · ∇_m ŷ‖ ≤ ε then
        t_{i_clip} ← Text Embedding from CLIP(e_i)
        sim_cosine ← r_clip · t_{i_clip}
        first_stage . append({t_i : sim_cosine}) ;        /* filtered tokens from ε neighborhood */
    end
end
{ t'_i } ← Top-20 tokens from first_stage sorted by sim_cosine in decreasing order ;   /* CLIP filtering */
for t'_i ∈ {t'_i} do
    e'_i ← GPT2Embedding( t'_i )
    if f(e'_i) = ŷ then
        explain_out . append(t'_i) ;                    /* Output preservation filtering */
    end
end
explain_out ← Top 4 tokens from explain_out sorted by log likelihood. ;     /* Final step */
```

---

## 4 EXPERIMENTS AND ANALYSIS

### 4.1 EXPERIMENTAL SETUP

Our proposed model was constructed using PyTorch, a Python-based deep-learning library. In our experimentation, we imported GPT2 from the Huggingface transformers package. All experiments were conducted on a single Nvidia A100 80GB GPU. We employed the Adam optimizer(Kingma & Ba, 2017) with a learning rate of $0.005$ for optimization. We use the Facebook Hateful Meme dataset(Kiela et al., 2021) for performing the experiments.

To ensure robust evaluation, we conducted a 5-fold cross-validation for testing after running experiments for $3,500$ steps on the respective train set. To mitigate the effects of non-determinism in GPU operations, we reported averaged scores obtained from 5 experiment runs. Additionally, we maintained a consistent random seed of $42$ across all our experiments.

---

[3]This condition is henceforth referred to as $\epsilon$-ball or $\epsilon$-neighborhood constraint interchangeably.

## 4.2 RESULTS

### 4.2.1 AUTOMATIC EVALUATION

For Automatic evaluation, we resort to using 'model faithfulness' as a guiding principle to evaluate the effect of the obtained keywords on model behavior. Especially, we measure 'simulatability', which can be defined as how well we can use the extracted keywords to predict the model's output. In Table 1, we depict the effect of various filtering mechanisms forming an ablation and compare our proposed method with the well-known input attribution-based methods, e.g. Integrated Gradient and KernelSHAP. For comparison, we use i) *Leakage adjusted simulatability (LAS) (Hase et al., 2020)* score: which measures the effect of predicted keywords/explanation on model prediction opting for explanation leakage. A positive LAS score reflects better 'simulatability'. For evaluating the effect of extracted keywords on model confidence, we use ii) *Comprehensiveness (↑)* and iii) *Sufficiency (↓)* metrics (DeYoung et al., 2020). We also list three accuracy-based measures: i) F1 score using both generated explanation and input meme as input to the model (denoted as **F1**), ii) F1 score using only input (denoted as **F1 w/ inp**) and iii) F1 score using only explanation (denoted as **F1 w/ exp**). We also propose two diversity-based metrics: i) Inter-sample diversity defined as Div (Inter) and ii) Intra-sample diversity defined as Div (Intra) elaborated in the Appendix Section D.

Table 1: *Automatic evaluation for faithfulness.* Empirical Performance of our proposed method in different setups (ablations) and comparison with baselines. **F1 w/ inp** is redundantly kept in the Table to aid easier comparison. 3rd-F and 4th-F refer to CLIP-based filtering step and output preservation steps, respectively.

| TopK | $\epsilon$-ball | 3rd-f | 4th-f | Div (Inter) | Div (Intra) | LAS (↑) | Compre. (↑) | Suff. (↓) | F1 | F1 w/ inp | F1 w/ exp |
|---|---|---|---|---|---|---|---|---|---|---|---|
| Random. | - | - | - | - | - | 0.0 | -2.42e-06 | 0.270 | 0.79 | 0.79 | 0.40 |
| Saliency Map | - | - | - | 4.06 | 7.43 | -0.004 | 0.004 | 0.140 | 0.79 | 0.79 | 0.68 |
| Inp x Grad (Shrikumar et al., 2017) | - | - | - | 3.25 | 7.40 | -0.004 | 0.005 | 0.148 | 0.79 | 0.79 | 0.69 |
| Int. Grad.(Sundararajan et al., 2017) | - | - | - | 3.62 | 6.94 | 0.000024 | 0.003 | 0.160 | 0.79 | 0.79 | 0.65 |
| KernelSHAP(Lundberg & Lee, 2017b) | - | - | - | 4.01 | 6.57 | 0.00052 | 0.002 | 0.180 | 0.79 | 0.79 | 0.59 |
| ✗ | ✗ | ✓ | ✓ | 10.81 | 3.81 | -0.0087 | 0.028 | 0.160 | 0.82 | 0.79 | 0.74 |
| ✗ | 0.05 | ✓ | ✓ | 6.28 | 6.94 | 0.0052 | 0.027 | 0.140 | 0.82 | 0.79 | 0.76 |
| ✗ | 0.01 | ✓ | ✓ | 5.96 | 7.47 | 0.01 | 0.035 | 0.133 | 0.83 | 0.79 | 0.76 |
| 3500 | ✗ | ✓ | ✓ | 5.53 | 7.21 | 0.02 | 0.043 | 0.079 | 0.84 | 0.79 | 0.84 |
| 2500 | ✗ | ✓ | ✓ | 5.48 | 7.16 | 0.023 | 0.042 | 0.077 | 0.84 | 0.79 | 0.85 |
| 1500 | ✗ | ✓ | ✓ | 5.23 | 7.17 | 0.016 | 0.038 | 0.077 | 0.83 | 0.79 | 0.85 |
| 500 | ✗ | ✓ | ✓ | 4.76 | 7.19 | 0.012 | 0.056 | **0.069** | 0.85 | 0.79 | 0.87 |
| 3500 | 0.1 | ✓ | ✓ | 5.53 | 7.19 | **0.027** | 0.043 | 0.085 | **0.85** | **0.79** | 0.85 |
| 2500 | 0.1 | ✓ | ✓ | 5.48 | 7.16 | 0.019 | 0.041 | 0.075 | 0.84 | 0.79 | 0.85 |
| 1500 | 0.1 | ✓ | ✓ | 5.24 | 7.17 | 0.019 | 0.038 | 0.076 | 0.83 | 0.79 | 0.85 |
| 500 | 0.1 | ✓ | ✓ | 4.75 | 7.15 | 0.023 | **0.056** | **0.071** | **0.85** | **0.79** | **0.88** |

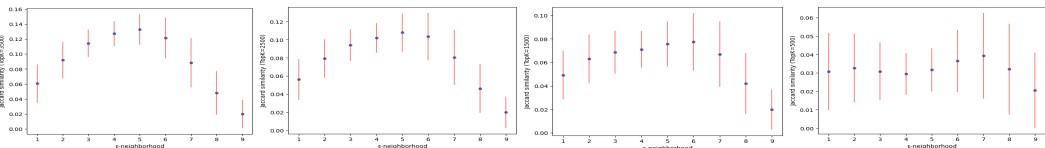

Figure 2: Plot of Jaccard similarity (Y-axis) between set of tokens in specific $\epsilon$ neighborhoods and for specific TopK sizes. Each plot refers to a specific TopK size. Each point in X-axis refers to an $\epsilon$ neighborhood bin starting from '$-0.09$ to $-0.07$' and ending at '$+0.07$ to $+0.09$'

**Comparison with baselines.** In the first five rows of Table 1, we describe the effect of extracting keywords from various input attribution-based explanation methods which are compared with random keywords. As expected, the random keywords obtained very low scores for all metrics reflecting the input attribution-based methods work well in practice. For every setup of our proposed model in the next 10 rows, we observe the superiority of our proposed approach by observing better obtaining scores for all the metrics. We also observe that although *F1 w/ exp* score is better for the baselines compared to the 'Random' explanations, the model performance remains the same when explanation and input both are used as input, as seen through the same F1 score obtained. This intuitively illustrates the fact that the extracted explanations do not provide extra information compared to inputs, such that the before and after F1 scores remain the same.

i) **Does epsilon ball constraint work in the absence of TopK constraint?** Firstly we consider dropping off the TopK sampling restriction (first stage) and observing the effect of disabling and

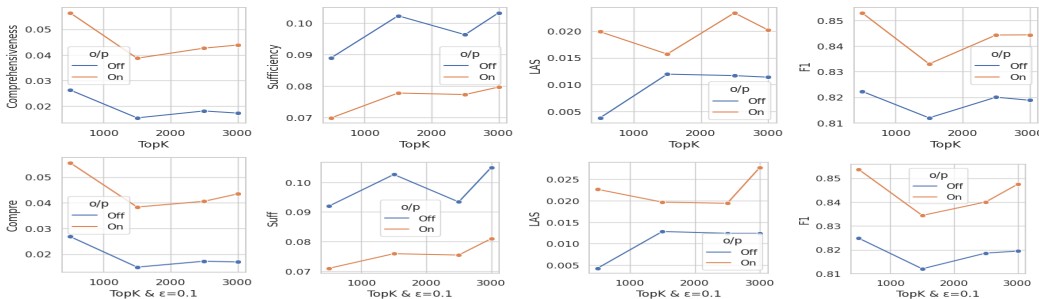

Figure 3: We plot values for various metrics for various fixed TopK values and $\epsilon$ values in X axis. On: O/P Preservation stage is on and Off: O/p preservation stage is off.

enabling the second stage with different values of $\epsilon$. Without any $\epsilon$ constraint, we obtain a negative LAS score along with a low 'comprehensiveness' score. It shows that only selecting the keywords using CLIP-based representation does not retrieve semantically relevant keywords. Next, we enable the second stage with two separate values of $\epsilon \in \{0.05, 0.01\}$. As can be seen through tabulated metrics, enabling the second stage has a positive effect on the quality of retrieved keywords. Also, $\epsilon = 0.01$ works better than $\epsilon = 0.05$ in terms of performance, suggesting that our theoretical justification of retrieving tokens in the neighborhood of $e^T \cdot \nabla_{\boldsymbol{m}} f(\boldsymbol{m}) \to 0$ indeed works well in practice.

ii) **Why would a larger TopK be associated with a lower comprehensiveness score?** From the Inter sample diversity score, we observe that a higher TopK value relates to higher Inter sample diversity, which entails that diversity between two explanation sets will be larger. Intuitively it can be seen that evaluating the model on a more diverse set leads to a lower probability of the predicted class due to lower model confidence. This consequently leads to lower comprehensiveness and higher sufficiency scores. Observe that there is a steady increase in inter-sample diversity with increasing TopK value, which further leads to lower comprehensiveness scores.

iii) **For the same TopK value what would the effect of enabling $\epsilon$ constraint be?** Comparing scores from the third and fourth rows, we observe that enabling the $\epsilon$ constraint seems to be beneficial for the 'simulatability', as can be seen by higher LAS scores for the same Top-K value without the $\epsilon$ constraint. This can be attributed to the same inter-sample diversity (indicating variations among samples) but lower intra-sample diversity (indicating lesser variation among retrieved keywords specific to an input meme). Less variation among the retrieved keywords for an input meme would intuitively mean better simulatability. However, this case is not always true, as a very low intra-sample diversity score would entail that the retrieved keywords are very similar in nature and would result in a low LAS score (observe the sixth row). Intuitively, there is a sweet spot where the ratio of inter-sample and intra-sample diversity would indicate optimally selected retrieved keywords.

iv) **What is the similarity of the retrieved explanations using a specific TopK value w.r.t various $\epsilon$ balls?** We observe that enabling the TopK constraint unequivocally retrieves better tokens as illustrated by Table 1. To theoretically justify it, we measure Jaccard similarity between the a) set of tokens retrieved using a specific Top-K value and b) tokens retrieved from the open $\epsilon$ neighborhood $[-\epsilon, +\epsilon]$. From Figure 2, we observe Jaccard similarity value spikes at $[-0.01, +0.01]$ when $TopK \in \{3500, 2500, 1500\}$ and at $[+0.02, +0.03]$ when $TopK \in \{500\}$. This entails Top-K retrieved tokens mostly lie in the neighborhood where $e^T \cdot \nabla_{\boldsymbol{m}} f(\boldsymbol{m}) \to 0$, which is theoretically justifiable.

v) **Is the Output preservation stage necessary?** We state that this stage is of utmost importance. From Figure 3, we observe that for different Top-K values (both with and without $\epsilon$ constraint enabled), enabling the output preservation stage always retrieves better keywords rather than disabling it. Also, the performance gap is quite significant, as seen in Figure 3.

vi) **Assessing the interpretive impact of keywords on model decisions.** We input extracted keywords as knowledge tokens during testing to empirically verify their influence on the model. The analysis reveals that in the majority of cases (86.23%), the predicted probability for the model-

Table 2: Human evaluation results based on 100 samples, evaluated by five annotators across Relevance and Exhaustiveness.

|  | Ours | $\epsilon$-ball (0.1) | CLIP | Integrated Gradient |
|---|---|---|---|---|
| Relevance | **3.26** | 2.64 | 2.65 | 2.43 |
| Exhaustiveness | **3.23** | 2.51 | 2.52 | 2.28 |

assigned class increased, with a mean rise of 0.06 and a standard deviation of 0.07. Moreover, the predicted class remains consistent in 97.41% of cases. While F1 score sees marginal increase of 0.39 %, this improvement, though not statistically significant, suggest the model's stable performance. Additionally, incorporating keywords does not adversely affect the model; instead, it bolsters confidence in predicting the class, emphasizing the interpretive value of keywords in shaping the model's decisions.

vii) **Is semantic relevance (CLIP filtering) stage necessary.** For every set-up in Table 1, we manually tested a random set of same 30 memes with and without the CLIP filtering stage enabled. Without CLIP filtering, the quality of the retrieved keywords are worse such that they do not semantically match with the input meme, which renders them unusable for end-use.

### 4.2.2 COMPARISON WITH MLLM

To assess the quality of keywords extracted by our framework in terms of *plausibility* and *interpretability*, we compare them with keywords extracted from LLaVa-1.5-13BLiu et al. (2023a) using manual annotation of 100 test set memes with ground truth keywords. Our method slightly outperforms LLaVa-1.5 in plausibility, as measured by bag-of-words cosine similarity with ground truth keywords. For interpretability, evaluated using the LAS score, LLaVa-1.5 performs slightly better than our model. However, the average scores for LAS and plausibility between LLaVa and our framework remain similar, indicating that our framework captures semantic nuances and interpretive capability comparable to an existing MLLM, despite having significantly fewer parameters (30X less). Additional details are provided in Appendix Section F.

### 4.2.3 ZERO-SHOT OOD GENERALIZATION

We assess the generalization capability of our models on HarMemePramanick et al. (2021) dataset in a zero-shot manner. This dataset contains out-of-distribution (COVID-19-related) memes when compared to our training dataset. Our method achieves a zero-shot Accuracy score of 64.91% which is better than random performance. The retrieved keywords are mostly explainable and can be adequately used to interpret the model. An example set of retrieved tokens can be seen in the Appendix Section G.

### 4.2.4 HUMAN EVALUATION

We perform a human evaluation of the generated keywords using two metrics, *viz.* Relatedness and Exhaustiveness (c.f. Appendix Section I). Relatedness is defined as how much a set of generated keywords is relevant to the content of the input meme, and Exhaustiveness is defined as how much of the aspects of an input meme are correctly represented in a set of retrieved keywords. Based on these definitions, five people (three authors of this paper and two students from the laboratory where the research is carried out) are chosen to rate the generated explanations (a randomly chosen set of 100 examples) on a scale of 1-5 in the 5 point Lickert scale. The inter-rater agreement (Cohen kappa score(Cohen, 1960)) for all the settings is more than 0.7, indicating fair agreement. For both exhaustiveness and relatedness, our methods achieve the best performance as observed from Table 2. Further, we observe an intuitive trend of the performance of the models, evaluated for both Relevance and Exhaustiveness as follows: Ours $>$ $\epsilon$-ball $\sim$ CLIP $>$ Integrated Gradient.

### 4.3 ANALYSING MODEL BEHAVIOR FROM KEYWORDS

In this section, we analyze the model qualitatively through the eyes of the extracted explainable keywords using various methods. For the correctly classified memes (with IDs 01276, and 98724), our proposed approach (TopK=3500 with $\epsilon = 0.1$ and other filters enabled) provided a relevant and

Table 3: Sample Qualitative analysis of our proposed method's output w.r.t several baseline outputs. Model outputs are shown for both success and failure cases for our model. Memes corresponding to the underlined IDs are shown in the Appendix Section H due to space constraints.

| | Meme | Ours | $\epsilon$-ball w/ CLIP | CLIP only | Integrated Gradient | Pred | Act |
|---|---|---|---|---|---|---|---|
| Correctly Classified | 01276 | philanthrop words encourage happiness | charitable encourage charity estimates | optimistic optimism worth Worth | smile worth thousand words | 0 | 0 |
| | 98724 | jews holocaust nazis hitler | nazis adolf sergei churchill | Adolf ologists Stalin Polish | eye loses bats normal | 1 | 1 |
| Misclassified | 91768 | jew holocaust hitler | jew abolished | jew jew Jew Jew | wearing :wtf normal adolf | 1 | 0 |
| | 13875 | cats cat lunch eat | cat cooperation sandwiches menu | cats cats cat Cat | see normal lunch let's | 0 | 1 |

exhaustive set of keywords for the input meme which can adequately represent the correct model prediction obtained. In fact, these explanations are also intuitive and help us to clarify that the model does not rely on any spurious correlation to predict its decision. For other variations of our proposed methods and the baseline method, we observe the quality of the retrieved keywords seems arbitrary with respect to the meme and model behavior. Thus they do not adequately reflect the reasoning based on which the model might have made its prediction. Even though the CLIP retrieves semantically relevant tokens, they are not exhaustive and often repetitive. This can even be seen from the very low intra-sample diversity score obtained by the CLIP-only method.

From meme ID 91768, we observe that the model predicts the meme as offensive even though it is a funny meme about Hitler. Due to the presence of Hitler's face, the model thinks of it as offensive, which is correctly illustrated by the retrieved keywords using our method. The baseline performs pretty poorly and the variations of our method give outputs that are either repetitive or not very semantically relevant to the input meme.

Another example is shown for meme Id 13875, where the model predicted an offensive meme as normal. The prediction appears to be influenced by the presence of the word 'cat,' which the model uses as a determining factor. This is a limitation of the model, as it lacks exposure to relevant memes during training, resulting in an inability to recognize the underlying issue of racism.

## 5 CONCLUSION

It is crucial to classify hateful content on social media platforms and generate explanations for moderation. Existing interpretability methods work on the input space, making it impossible to generate correct explanations for contents with hidden meanings. Our work allows us to not only judge if the shared content is hateful and to be moderated but also gives an explanation that might be absent in the input. Our work allows us to find out the hidden meaning behind a meme more efficiently and also helps us elucidate the model's decision-making process. It is also helpful to find any model bias as shown in the qualitative evaluation. Our proposed method surpasses all the other available models by both automated and manual evaluation. It can be used on social media platforms and is expected to have a huge real-life impact. Further, our method is designed to be task agnostic, and it can be extended to generate good explanations for other decision-making tasks. It is out of the scope of this study to show the effect of our proposed approach beyond the domain of memes due to space limitations. Subsequently, to facilitate the robustness of the proposed approach, we aim to show its performance in various visual-linguistic tasks (*viz.* Visual Question Answering, Visual NLI etc.) as a part of future studies.

ETHICS STATEMENT

The dataset used in this paper is publicly available. Our resources used in this paper were derived from publicly available memes, and we strictly adhered to the regulations regarding data usage to avoid any infringement of copyright laws. Moreover, our study underwent evaluation and approval by our Institutional Review Board (IRB). The memes in the dataset may contain offensive keywords, and we thus advise the reader of the paper to exercise discretion while using the memes. Overall our paper is all about uplifting the online community by understanding offensive memes better.

REPRODUCIBILITY STATEMENT

We make all our results reproducible by using a random seed of $42$ throughout the paper. The codes are available here: `https://anonymous.4open.science/r/Expleme-E8BE/`. The outputs obtained from the model will be provided upon the acceptance of the paper and can be cross-checked by the outputs obtained from the code. Our method is also theoretically justifiable as can be seen from the proof of the proposed theorem and the Proposition in Appendix section B. The assumptions are clearly mentioned based on which the theorems were proved.

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

# A  PROOF

**Proof.** We have $\hat{y} = f(\boldsymbol{m})$ and $\boldsymbol{m}^+ = \boldsymbol{m} + t\nabla_{\boldsymbol{m}}\hat{y}$ for $t > 0$ and $t$ is a scalar, called step size. As $f(\boldsymbol{m})$ is strongly convex,

$$f(\boldsymbol{m}^+) \geq f(\boldsymbol{m}) + \nabla_{\boldsymbol{m}}{}^T\hat{y}(\boldsymbol{m}^+ - \boldsymbol{m}) + \epsilon\|\boldsymbol{m}^+ - \boldsymbol{m}\|^2 \tag{1}$$

where $\epsilon > 0$. So we get,

$$f(\boldsymbol{m}^+) \geq f(\boldsymbol{m}) + (t + \epsilon t^2)\|\nabla_{\boldsymbol{m}}{}^T\hat{y}\|^2 \tag{2}$$

As $t + \epsilon t^2 > 0$, so

$$f(\boldsymbol{m}^+) \geq f(\boldsymbol{m}) + K\|\nabla_{\boldsymbol{m}}{}^T\hat{y}\|^2 \tag{3}$$

where $K$ is a positive real number.

By Cauchy-Schwartz inequality, assuming $e^T$ and $\nabla_{\boldsymbol{m}}\hat{y}$ are not aligned,

$$e^T.(\nabla_{\boldsymbol{m}}\hat{y}) < \|e\|\|\nabla_{\boldsymbol{m}}{}^T\hat{y}\| \tag{4}$$

So, by Equation 3 and 4 and assuming $e^T$ and $\nabla_{\boldsymbol{m}}\hat{y}$ are not orthogonal,

$$e^T.\nabla_{\boldsymbol{m}}f(\boldsymbol{m}^+) > e^T.\nabla_{\boldsymbol{m}}f(\boldsymbol{m}) + Ke^T.\nabla_{\boldsymbol{m}}(\frac{e^T.\nabla_{\boldsymbol{m}}\hat{y}}{\|e\|})^2 \tag{5}$$

$$e^T.\nabla_{\boldsymbol{m}}f(\boldsymbol{m}^+) > e^T.\nabla_{\boldsymbol{m}}f(\boldsymbol{m}) + 2K(e^T.\nabla_{\boldsymbol{m}}f(m))e^T.\nabla_{\boldsymbol{m}}(e^T.\nabla_{\boldsymbol{m}}f(m)) \tag{6}$$

$$e^T.\nabla_{\boldsymbol{m}}f(\boldsymbol{m}^+) > e^T.\nabla_{\boldsymbol{m}}f(\boldsymbol{m})(1 + 2Ke^T.\nabla_{\boldsymbol{m}}(e^T.\nabla_{\boldsymbol{m}}f(m))) \tag{7}$$

Taking gradient w.r.t $\boldsymbol{m}$ and making sure the inequality still holds true come from the Proposition in Appendix Section B.

Also, by the definition of Hessian Matrix $\boldsymbol{H}(f)(\boldsymbol{m})$ of $f(m)$, we can write,

$$e^T.\nabla_{\boldsymbol{m}}(e^T.\nabla_{\boldsymbol{m}}f(m)) = \boldsymbol{e}^T.\boldsymbol{H}(f)(\boldsymbol{m}).\boldsymbol{e}$$

(The above identity can be seen true by expansion of terms)

As f(m) is strongly convex, we know its Hessian $\boldsymbol{H}(f)(\boldsymbol{m})$ at any point is positive definite.

Also by the definition of positive semi-definiteness, $\boldsymbol{a}^T.\boldsymbol{H}(f)(\boldsymbol{z}).\boldsymbol{a} > 0$, where $\boldsymbol{a}$ are non-zero real vectors. Considering $\boldsymbol{a}$ as $\boldsymbol{e}$ as we assumed $\boldsymbol{e}$ are non-zero vectors, we can write $\boldsymbol{e}^T.\boldsymbol{H}(f)(\boldsymbol{m}).\boldsymbol{e} > 0$, which implies, $e^T.\nabla_{\boldsymbol{m}}(e^T.\nabla_{\boldsymbol{m}}f(m)) > 0$

From the above argument, and from Equation 7,

$$e^T.\nabla_{\boldsymbol{m}}f(\boldsymbol{m}^+) > e^T.\nabla_{\boldsymbol{m}}f(\boldsymbol{m})\rho \tag{8}$$

where $\rho$ is always greater than 1.

# B PROPOSITION

We assume the following parameters regarding the step size. The step size is small such that $t \to 0$. This entails $\boldsymbol{m}^+ \to \boldsymbol{m}$.

Also,

$$h(\boldsymbol{m}) = f(\boldsymbol{m}^+) - f(\boldsymbol{m}) - K\left(\frac{e^T.\nabla_{\boldsymbol{m}}\hat{y}}{\|e\|}\right)^2 \tag{9}$$

Now for $\boldsymbol{m}^+ > \boldsymbol{m}$

$$\delta h = h(\boldsymbol{m}^+) - h(\boldsymbol{m}) =$$

$$f(\boldsymbol{m}^{++}) - f(\boldsymbol{m}^+) - K\left(\frac{e^T.\nabla_{\boldsymbol{m}}f(\boldsymbol{m}^+)}{\|e\|}\right)^2 - \left(f(\boldsymbol{m}^+) - f(\boldsymbol{m}) - K\left(\frac{e^T.\nabla_{\boldsymbol{m}}f(\boldsymbol{m})}{\|e\|}\right)^2\right) \tag{10}$$

We know $K > 0$ Also, $\left(\frac{e^T.\nabla_{\boldsymbol{m}}f(\boldsymbol{m}^+)}{\|e\|}\right)^2 > 0$ and $\left(\frac{e^T.\nabla_{\boldsymbol{m}}f(\boldsymbol{m})}{\|e\|}\right)^2 > 0$ and we also assume that the alignment between $e$ and $\nabla_{\boldsymbol{m}}f(\boldsymbol{m}^+)$ is mostly similar to the alignment between $e$ and $\nabla_{\boldsymbol{m}}f(\boldsymbol{m})$ as we previously assumed $\boldsymbol{m}^+ \to \boldsymbol{m}$. So,

$$\left(\frac{e^T.\nabla_{\boldsymbol{m}}f(\boldsymbol{m})}{\|e\|}\right)^2 - \left(\frac{e^T.\nabla_{\boldsymbol{m}}f(\boldsymbol{m}^+)}{\|e\|}\right)^2 \approx 0$$

Notationally, we denote (assume), $\nu = \nabla_m f(\boldsymbol{m})$ and, $\boldsymbol{m} \to \boldsymbol{m}^+$ further entails $\nu \to \nu^+$.

$$\delta h \approx f(\boldsymbol{m}^{++}) - f(\boldsymbol{m}^+) - (f(\boldsymbol{m}^+) - f(\boldsymbol{m}))$$

$$\delta h = \frac{t^2}{t^2}\delta h$$

As, $\boldsymbol{m}^+ = \boldsymbol{m} + t\nabla_m f(m)$ and we act at the regime where $t \to 0$,

$$\delta h = \frac{t^2}{t}\lim_{t \to 0}\left(\frac{f(\boldsymbol{m}^+ + t\nu^+) - f(\boldsymbol{m}^+)}{t} - \frac{(f(\boldsymbol{m} + t\nu) - f(\boldsymbol{m}))}{t}\right)$$

$$\delta h = \frac{t^2}{t}\lim_{t \to 0}(f'(\boldsymbol{m}^+; \nu^+) - f'(\boldsymbol{m}; \nu))$$

As $\nu^+ \to \nu$, when $t \to 0$

$$\delta h = t^2 \lim_{t \to 0}\frac{f'(\boldsymbol{m} + t\nu; \nu) - f'(\boldsymbol{m}; \nu)}{t}$$

By the definition of directional derivative of $f(\boldsymbol{m})$ in the direction of $\nu$,

$$\delta h = t^2 \lim_{t \to 0}\frac{f'(\boldsymbol{m} + t\nu).\nu - f'(\boldsymbol{m}).\nu}{t}$$

$$\delta h = t^2 \lim_{t \to 0}\frac{\nu_i \partial_{\boldsymbol{m}_i} f(\boldsymbol{m} + t\nu) - \nu_i \partial_{\boldsymbol{m}_i} f(\boldsymbol{m})}{t}$$

$$\delta h = t^2 \nu_i \partial_{m_i m_j} f(\boldsymbol{m})\nu_j$$

$$\delta h = t^2 \nu^T \boldsymbol{H}(f)(\boldsymbol{m})\nu$$

As f(m) is strongly convex, we know its Hessian $\boldsymbol{H}(f)(\boldsymbol{m})$ at any point is positive definite.

Also by the definition of positive semi-definiteness, $\boldsymbol{a}^T.\boldsymbol{H}(f)(\boldsymbol{z}).\boldsymbol{a} > 0$, where $\boldsymbol{a}$ are non-zero real vectors. Considering $\boldsymbol{a}$ as $e$ as we assumed $e$ are non-zero vectors, we can write $\nu^T.\boldsymbol{H}(f)(\boldsymbol{m}).\nu > 0$. Also $t > 0$. So by definition $\delta h > 0$. As $h$ is an increasing function of $\boldsymbol{m}$ so, $\nabla_{\boldsymbol{m}}h(\boldsymbol{m}) > 0$.

### B.1 COROLLARY

If we assume

$$h(\boldsymbol{m}) = f(\boldsymbol{m}^+) - f(\boldsymbol{m})$$

, then by a similar argument, without choosing the assumption that

$$(\frac{e^T.\nabla_{\boldsymbol{m}}f(\boldsymbol{m})}{\|e\|})^2 - (\frac{e^T.\nabla_{\boldsymbol{m}}f(\boldsymbol{m}^+)}{\|e\|})^2 \approx 0$$

We can write $\nabla_{\boldsymbol{m}}h(\boldsymbol{m}) > 0$. Which entails $\nabla_{\boldsymbol{m}}f(\boldsymbol{m}^+) > \nabla_{\boldsymbol{m}}f(\boldsymbol{m})$. This inequality further entails $e^T.\nabla_{\boldsymbol{m}}f(\boldsymbol{m}^+) > e^T.\nabla_{\boldsymbol{m}}f(\boldsymbol{m})$, where $\boldsymbol{e}$ must be non-zero and non-negative real column vectors. In turn, this would entail that if $e^T.\nabla_{\boldsymbol{m}}f(\boldsymbol{m}) \to 0$, then $e^T.\nabla_{\boldsymbol{m}}f(\boldsymbol{m}^+) > 0$, which signifies the **alignment vs optimization tradeoff**.

## C SYSTEM DESIGN

Our proposed system is comprised of two subsystems, defined by one multimodal encoder and another GPT2 as the classifier as depicted by Figure 4. Our proposed method works in three stages: i) Multimodal encoding via CLIP, ii) Generating differentiable approximation of the label, iii) Preparing input for GPT2.

### C.1 STAGE 1: MULTIMODAL ENCODING

The first stage involves encoding the input meme and embedded text using the CLIP module, resulting in separate text and image embeddings. These embeddings are combined through a projection layer and subsequent average-pooling, yielding a multimodal embedding.

### C.2 STAGE 2: CLASSIFICATION AND LABELING

In the second stage, the multimodal embedding is processed using an FFN layer to classify the input meme, generating a corresponding label. The label then undergoes a Gumbel-Softmax layer, producing a one-hot representation while preserving gradient information. Additionally, the GPT2 embedding of the verbalized label is obtained by passing it through the GPT2 embedding layer. The final step involves element-wise multiplication of the GPT2 embedding with the one-hot representation, resulting in an embedding of the output label with the original gradient information.

### C.3 STAGE 3: END-TO-END TRAINING

The third stage integrates the multimodal embedding from Stage 1 and label information from Stage 2, along with GPT2 embeddings of knowledge text, meme text, and meme caption (indicated by [CAP] due to space constraints), and the remaining prompt. This composite input is provided to the GPT2 language model. Subsequently, both the GPT2 and the FFN from Stage 2 undergo end-to-end training.

For a visual representation of the overall system architecture, please refer to Figure 4 in the main text.

## D PROPOSED METRICS

We define two metrics for measuring the diversity of the generated samples for a particular meme (referred to as Intra-sample diversity) and for the whole test set (referred to as inter-sample diversity). Intra-sample diversity metric is used to measure how diverse a set of retrieved tokens is. As an example, let us suppose we have the following four keywords retrieved for a particular meme: {*Hitler, Adolf, Jews, WW2*}. Although the set of keywords correctly reflects the meme is related to WW2 and nazi Germany, it is less diverse than another set: {*Nazi, Antisemitism, Holocaust, Hitler*} where the collected words are more exhaustive and diverse. High diversity would mean that the generated keywords are not related to each other and thus may not refer to a specific topic. So high

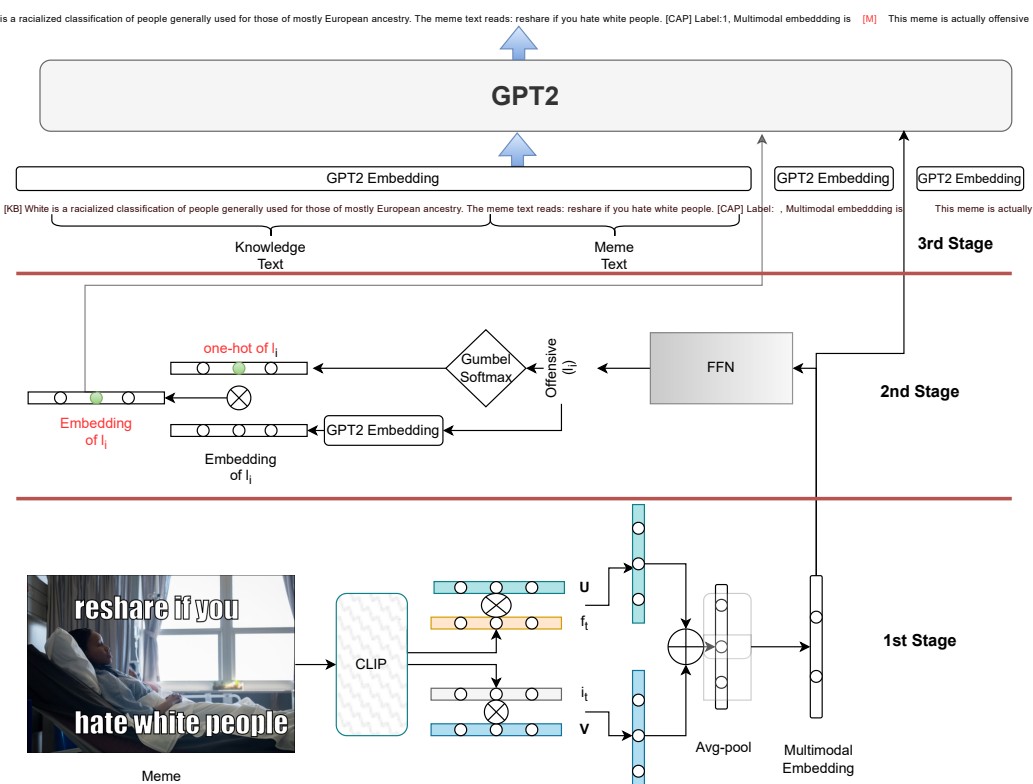

Figure 4: Diagram outlining our proposed system with three key stages. In the first stage, CLIP embedding generates a multimodal embedding ($M_t$) through a sum-pool operation. The second stage involves classifying $M_t$ using an FFN, followed by a Gumbel Softmax and GPT2 embedding layer. This results in a differentiable embedding representation of the predicted output from the FFN. Red color text denotes that the vector is equipped with its gradient information. The third stage incorporates GPT2 embedding layers for both knowledge text and meme text, along with the differentiable embedding representation from the second stage and $M_t$. Ultimately, this stage reconstructs label information using GPT2. At the inference stage, we generate explainable keywords by sampling from this GPT2 model given the prompt.

diversity is not desirable. Similarly, too low diversity would also be undesirable because it entails the repetitive nature of the retrieved keywords.

**Intra-sample diversity**. Mathematically we first calculate the word vectors ($\boldsymbol{v}_i \in S$) inside a sample $S$ for the word $w_i$ by GLoVe. The intra-sample diversity score ($i1$) is then defined as:

$$i_1 = \frac{1}{N} \sum_{i=1}^{N} \|\boldsymbol{v}_i - \mu(\boldsymbol{v}_i)\|_2 \tag{11}$$

, where $\mu(\boldsymbol{v}_i)$ is the mean of the word vectors defined as $\mu(\boldsymbol{v}_i) = \frac{1}{N} \sum_{i=1}^{N} \boldsymbol{v}_i$ and $N$ is the number of samples (typically $N = 4$) retrieved for one particular meme.

**Inter-sample diversity**. This is a dataset-wide metric. We measure how similar or dissimilar (on average) two samples (a sample refers to $N$ retrieved keywords specific to an input meme) are. This is defined similarly to Intra-sample diversity as:

$$i_2 = \frac{1}{M} \sum_{i=1}^{N} \|\mu(\boldsymbol{v}_i) - \psi(\mu(\boldsymbol{v}_i))\|_2 \tag{12}$$

, where $\psi(\boldsymbol{v}_i) = \frac{1}{M} \sum_{i=1}^{N} \boldsymbol{v}_i$ and $M$ is the number of samples in the dataset.

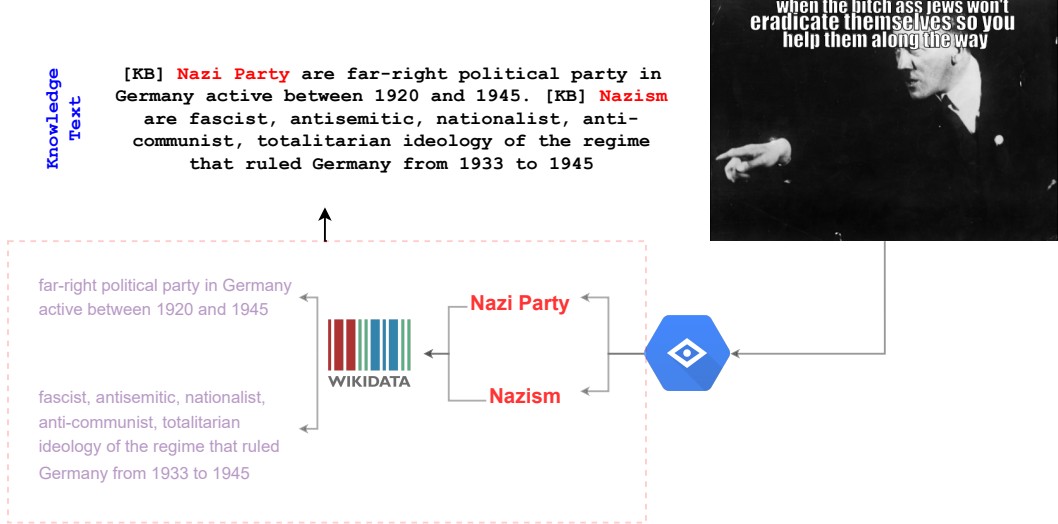

Figure 5: Procedure to extract the knowledge text which is subsequently used as a prompt in the GPT2 model in our proposed system.

Table 4: Plausibility (measured by cosine similarity) and Interpretability (measured by LAS) for Llava extracted keywords compared to extracted keywords from our framework.

| Metrics | LlaVa | Ours |
|---|---|---|
| Cosine Sim | 0.56 | **0.64** |
| LAS | **0.15** | 0.09 |

## E  EXTRACTION OF KNOWLEDGE

Our proposed model leverages discrete knowledge obtained through sources, such as Wiki-Data(Vrandečić & Krötzsch, 2014) and the Google Cloud Vision API. To formally describe our approach, we begin by retrieving a set of n keywords ($K = \{k_1, k_2, ..., k_n\}$).

For each keyword $k_i$, we conduct a search in WikiData to find a corresponding description, denoted as $d_i$. These pairs of keywords and their associated descriptions ($k_i$ and $d_i$) are then combined to create knowledge texts ($kt_i$), which take the following structured form: "**[KB]** $k_1$ are $d_1$ .... **[KB]** $k_n$ are $d_n$, **[CAPTION]** $c_i$". Here, the placeholders [KB] and [CAPTION] serve as special tags to represent the $k_i$, $d_i$ pairs, and the meme's caption ($c_i$) respectively. The term are functions as a conjunction between each $k_i$, $d_i$ pair, ensuring the formation of coherent sentences. Captions ($c_i$) are generated using the OFA module(Wang et al., 2022). Figure 5 shows the process step-by-step.

## F  COMPARISON TO LLAVA

To compare fairly against MLLM (specifically Llava), we resort to the following steps:

We give a meme to Llava and ask it to generate whether it is offensive or not by giving it the prompt: "Is this meme offensive?" The output in natural language is followed by another question: "Give four one-word keywords that summarize your explanation." which outputs four keywords just like our model.

Subsequently, we compare the LAS score of the Llava-produced output to that of our proposed model. A higher LAS score obtained by our model reflects higher faithfulness and thus higher model interpretability. (obviously, the explanation in Natural Language is bound to be better than our proposed method of retrieving contextual keywords but it is not fair to compare an NL output to that of a set of keywords; therefore the 2nd step is done to extract relevant keywords from LlaVA).

Table 5: Example memes from HarMeme dataset. We use our model in a zero-shot manner to extract the salient keywords using our proposed framework.

| Meme ID | Ours |
|---------|------|
| 2067 | disorder, clinic, effect, clinically |
| 1516 | socialist, criminals, economic |
| 5425 | hero, villain, horror, alien |
| 5582 | refugee, america, forced, presidential |
| 5595 | revealed, autistic, legitimate, diagnostic |
| 5528 | genocide, situation, acting, considered |
| 5649 | comedy, triggered, crisis, liberals |
| 0581 | responsible, widespread, delusional, genetic |

Table 6: Sample keywords retrieved from LlaVa and our method for sample memes.

| IDs | GT | LlaVa-1.5-13B | Ours |
|-----|-----|---------------|------|
| 01924 | oppression, racism, anti-feminism, suicide-bombing | Offensive, Violent, Comparison, Women's rights | racism, oppression, slavery, feminists |
| 96328 | racism, anti-white, derogatory, hospital | racism, discrimination, harmful, offensive | whites, bigot, refugee, racism |
| 39862 | anti-protestanism, inability, mocking, sarcasm | Offensive, disrespectful, insensitive, inappropriate | pedestrians, ter, pro, logic |

Additionally, we pick 100 memes and manually annotate each one of them with a set of four keywords. We measure cosine similarity between the bag-of-word of set A (retrieved from our model) and Ground truth set (let's call the average cosine similarity for this case for 100 memes as Cossim-ours). Similarly, we get the average cosine similarity between the Llava set and ground truth for 100 memes (we call it Cossim-llava). Cossim-ours > Cossim-llava shows that our method fairs better in terms of plausibility.

So, in summary when considering the richness of extracted keywords (not NL explanation, as this would not be fair and it is out of the scope of our paper to compare against verbalized NL outputs), our method fairs better for both faithfulness (i.e interpretability) and plausibility (i.e. explainability) that state-of-the-art MLLM models like LlaVa.

Table 4 describes the value of cosine similarity (a measure of explainability) obtained for LlaVa and our method and it also shows the obtained LAS score (a measure of interpretability). Some extracted keywords are shown in Table 6. The relevant memes are shown in Figure 8.

## G  ZERO SHOT PERFORMANCE IN HARMEME DATASET

Some example data points are shown in Table 5 for which we show the interpretable keywords extracted from our proposed framework. The corresponding memes can be seen in Figure 7. We observe that for a particular meme, the corresponding keywords adequately describe the meme.

## H  QUALITATIVE ANALYSIS OF MEMES

The relevant memes related to the qualitative analysis are shown here due to space constraints.

## I  SCALE OF RELATEDNESS AND EXHAUSTIVENESS

**Relatedness** is defined as how much a set of generated keywords is relevant to the content of the input meme, i.e. if all generated keywords are relevant to the meme, the score is five, and if none of them are related, the score is one. The scores between two and four are given when some of the keywords are relevant or partially relevant. In-between scores are subjectively rated by the evaluators depending on their understanding.

**Exhaustiveness** is defined as the amount of coverage of the theme of a meme through keywords. If the meme can be completely explained with the generated keywords, the score should be five, and if

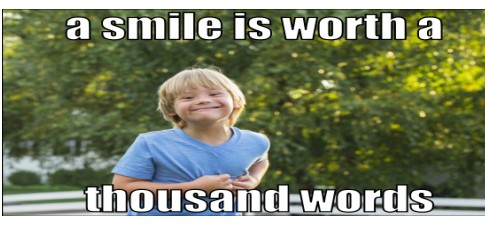
01276

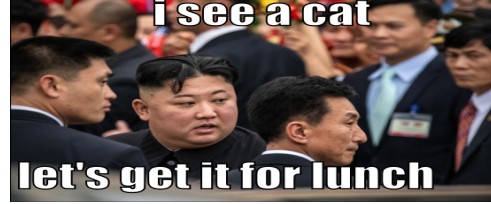
13875

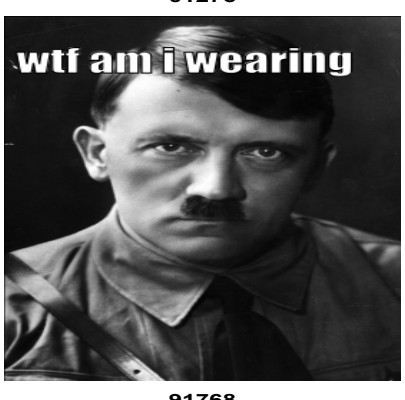
91768

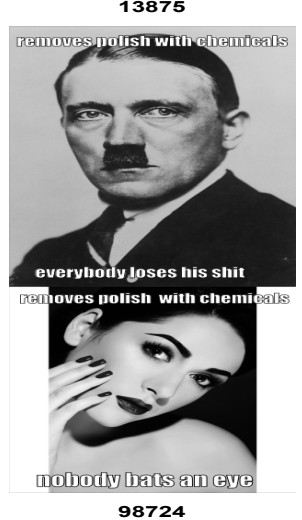
98724

Figure 6: Related memes for qualitative analysis

the generated keywords are unable to convey any meaningful information about the meme and are insufficient to explain it, the score is one.

## J    LIMITATION

**1.    Task Agnosticism and Generalizability:**  While our research paper presents a task-agnostic method, it is crucial to acknowledge that the extent of its task-agnostic nature remains to be fully explored.  The current study focuses on a specific task, and the generalizability of the proposed method to a broader range of tasks is an open question.  Future research should investigate the applicability and effectiveness of the proposed approach across diverse tasks to establish its true task-agnostic capabilities.

**2.    Simple Keywords and Natural Language Output:** The paper employs simple keywords for interpreting model decision.  While these keywords provide a foundational understanding, more in-depth exploration is necessary to refine the language and generate *precise natural language sentences* that effectively convey the nuances of the proposed model and the input meme used. Subsequent work should involve a detailed analysis and improvement of the language used, ensuring that the conveyed message is accurate, clear, and well-structured in the form of a grammatically correct sentence.

## K    MODALITY IMPORTANCE

Meme classification is multimodal task and it is essential to quantify importance of each modality for the corresponding explanation generated from our model.  We qualitatively analyse four memes where our framework is compared to text-only and image-only baselines.  These baselines are formed by only passing either textual or visual representation from the CLIP to the downstream

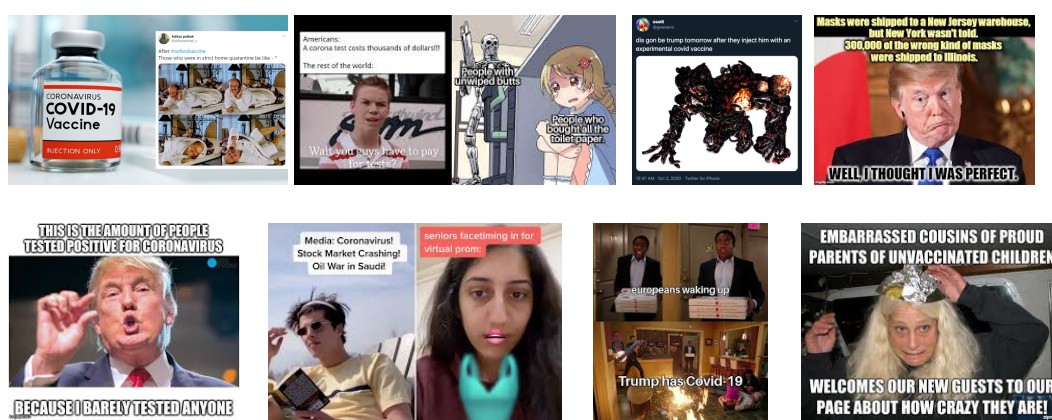

Figure 7: Sample memes corresponding to the meme ids in Table 5. Memes are sorted from left to right, top to bottom according to their IDs.

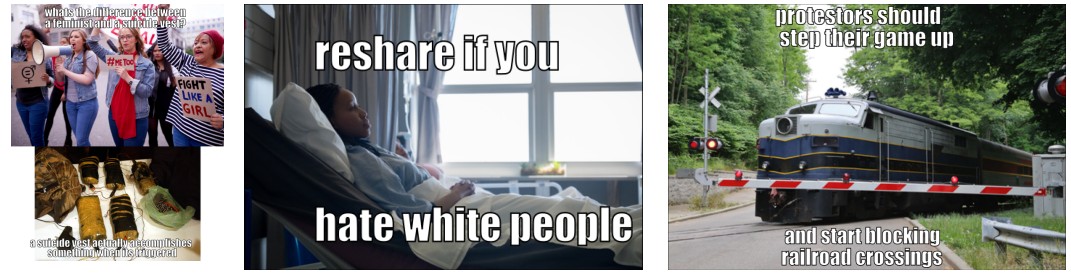

Figure 8: Sample memes corresponding to the meme ids in Table 6. Memes are sorted from left to right according to their IDs in Table.

pooling layer. See Section 3 for more details. From the qualitative analysis (Table 7) the following things can be inferred:

1. Multimodal (Ours) representation achieves the best quality of retrieved keywords as expected. Though it is closely rivalled by text only representation, for the last two memes (Id: 91768 an 13875), where our method fails to classify the offensiveness class, the text-only baselines perform pretty poorly too. Even the extracted keywords do not make much sense.

2. Image only baselines perform pretty poorly. The performance is even poorer that text only baselines. This is probably due to how the CLIP based filtering stage is designed, where the image part of the meme is compared to the text part of the meme. This downstream process might lead to non-sensical output when visual representation is used as the unimodal counterpart of textual representation.

3. We tabulate the classification performance of these baselines in Table 8. As per with our intuition, multimodal learning beats both unimodal (both textual and visual) representation based baselines.

Table 7: Sample Qualitative analysis of our proposed method's output w.r.t unimodal baselines. Memes corresponding to the underlined IDs are shown in the Appendix Section H.

| Meme | Ours | Text-only | Image-only |
|---|---|---|---|
| 01276 | *philanthrop words encourage happiness* | *philanthrop words encourage happiness* | *philanthrop words encourage happiness* |
| 98724 | *jews holocaust nazis hitler* | *nazis adolf holocaust jews* | *Adolf jew holocaust nazis* |
| 91768 | *jew holocaust hitler* | *die ger dictator propaganda* | *jew holocaust hitler german* |
| 13875 | *cats cat lunch eat* | *cats cat lunch sat* | *none* |

Table 8: Unimodal and Multimodal performance of our proposed method for the task of offensiveness detection. Note we report final classification result as obtained from the LLM, not from the intermediate classifier layer.

| Metrics | Text | Vision | Multimodal |
|---|---|---|---|
| Acc | 70.29 | 66.35 | 75.64 |
| F1 | 67.77 | 64.38 | 73.46 |

