# OpenReview forum: "EXPLEME: A Study in Meme Interpretability, Diving Beyond Input Attribution"
_ICLR.cc/2024/Conference — ICLR 2024 Conference Withdrawn Submission_

### Official Review · Reviewer_Diur · 2023-10-23

**Soundness:** 2 fair
**Presentation:** 2 fair
**Contribution:** 2 fair
**Rating:** 3
**Confidence:** 4

**Summary:**

The paper proposes an explanation approach for memes, which provides a more rich set of relevant keywords with the goal of providing richer background information about the semantics of an input meme. The proposed approach is leveraging multimodal encoding and classification through a Language Model (GPT2) with the goal of generating a set of supporting keywords that are not necessarily appearing in the input image, but are inferred (with the help of GPT2) to be relevant to the input meme. Quantitative and qualitative evaluation provides evidence in favour of the good performance of the proposed approach.

**Strengths:**

- Motivation and problem formulation are well described.
- Several of the proposed idea elements, e.g. prediction preservation, are quite original and well integrated in the overall framework.
- The experimental results appear promising.

**Weaknesses:**

- There are certain major methodological issues (cf. Questions).
- The related work coverage is insufficient.
- The quality of the manuscript is below publication standards.

**Questions:**

The motivation behind this work is clear and very relevant. However, a high-level description/explanation of the inner workings of the proposed method is missing from the introduction.

The literature review of multi-modal hate speech detection is rather poor, comprising all in all 9 lines, with half of them being about text-based detection.

The proposed idea is quite similar to concept bottleneck models; the authors could at least add a reference and discuss the differences between the two approaches.

The quality of writing requires considerable improvement.

The methodology presentation is rather unclear and high-level, which makes it very difficult to understand the details of the implementation and to connect the pieces.

To me there are a couple of methodological flows:
1) The fact that GPT predicts certain keywords (based on the image, text, image-caption features & prediction embeddings) does not serve as an explanation of how the two modalities interact in order to produce hate or not. It just automatically provides the context around the meme. Additionally, based on own experience and experiments with OFA to caption memes, the result is often bad, most of the times unsuccessfully trying to OCR the meme instead of providing a good caption about the background.
2) The biases of GPT model are not alleviated of accounted for in this analysis. It is used as if it was perfect.
3) In general, a set of keywords is hardly sufficient to capture the semantic nuance that most memes carry. In the age of LLMs, one would expect an explainability approach to rely on some more sophisticated generative method to produce explanations that are readable.

The authors claim that the method is applicable to other contexts/tasks but give no evidence of that. Therefore, I feel they should remove this from the list of claimed contributions.

---

> ### Author Response · Authors · 2023-11-19
> **Response to questions from Reviewer Diur (part 1)**
>
> Questions:
>
> > Q1. The motivation behind this work is clear and very relevant. However, a high-level description/explanation of the inner workings of the proposed method is missing from the introduction
>
> Thank you for your constructive feedback. In response, we have incorporated a succinct paragraph into the introduction to address space constraints. Additionally, we have refined the depiction of the proposed model for improved clarity, as presented in Appendix Figure 4 and its corresponding caption. We elaborate the high level details in Appendix section C (System Design).
>
>
> > Q2. The literature review of multi-modal hate speech detection is rather poor, comprising all in all 9 lines, with half of them being about text-based detection.
>
> We have revised the literature review, incorporating additional references that align more closely with the context of the paper.
>
> > Q3. The proposed idea is quite similar to concept bottleneck models; the authors could at least add a reference and discuss the differences between the two approaches.
>
> The concept bottleneck operates on the principle that discrete concepts can be introduced through constraints at the hidden layer of a neural model. In this supervised approach, distinct concept classes are required for each input instance. It's important to note that our proposed idea does not involve a hidden layer where the concept bottleneck model explicitly enforces the infusion of discrete concept classes. As far as our understanding goes, there is no overlap between these two methods, and they appear to be completely disjoint. Please feel free to seek further clarification if needed.
>
> > Q4. The quality of writing requires considerable improvement.
>
> We apologize for any inconveniences that might have caused you in the revised version, we have tried to improve it in the best possible way. However if you have any specific suggestion to improve any specific sections, we will be incorporating that.
>
> > Q5. The methodology presentation is rather unclear and high-level, which makes it very difficult to understand the details of the implementation and to connect the pieces.
>
> Complementary to the high-level methodology presentation, we update the figure of the model (Appendix Figure 4) along with a long and suitable caption to illustrate the low level details.
>
> > Q6. The fact that GPT predicts certain keywords (based on the image, text, image-caption features & prediction embeddings) does not serve as an explanation of how the two modalities interact in order to produce hate or not. It just automatically provides the context around the meme. Additionally, based on own experience and experiments with OFA to caption memes, the result is often bad, most of the times unsuccessfully trying to OCR the meme instead of providing a good caption about the background.
>
> First part: Indeed, the explanations generated by our framework not only offer contextual information around the meme but also provide interpretable tokens crucial for understanding the model's decision-making process. To substantiate this claim, we conducted an empirical experiment for validation. During testing: i) Initially, we assessed the model's confidence in the predicted class, and ii) Subsequently, we introduced knowledge tokens comprising the top 4 retrieved keywords from our framework. If these tokens genuinely interpret the model's inner workings, confidence in the initially predicted class should increase. Our empirical findings reveal a positive numerical difference between the two measures for 97.41% of the samples. For an in-depth discussion, please refer to section 4.2.1 point VI in the updated paper.
>
> Second part: There might be a misunderstanding in this context. OFA is utilized as the captioning module responsible for generating captions for the memes. These input captions are not wrong but might not always contain adequate information. That is the reason (to increase adequate information contained in the caption) we also use external knowledge tokens retrieved from Google Cloud Vision API and WikiData (c.f. Appendix Section E). These captions along with the external knowledge are then fed into the GPT model. It's important to note that OFA's role is in generating captions and does not influence how GPT predicts its output. For example, Consider a meme about Donald Trump in the training set (meme id:90126), the OFA caption is: “the picture of a man standing beside the American flag”. This caption, although correct, does not have the information that the man might be Donald Trump. This is where external knowledge comes into play. The external knowledge corresponding to this meme is: “[KB] Donald Trump are president of the United States from 2017 to 2021.” The OFA caption and the knowledge text are then combined and forwarded to the model.

---

> > ### Author Response · Authors · 2023-11-19
> > **Response to questions from Reviewer Diur (part 2)**
> >
> > > Q7. The biases of GPT model are not alleviated of accounted for in this analysis. It is used as if it was perfect.
> >
> > Thanks for the suggestion. While the GPT model serves as the final classifier, it also functions as the module employed during inference to extract interpretable keywords related to the predicted class from the model. It is accurate that the quality of the extracted keywords may be affected by any bias present in GPT. However, the model is utilized off-the-shelf to showcase its effectiveness in extracting meaningful keywords, not for the purpose of detecting and mitigating bias. It's worth noting that the extracted keywords could potentially be valuable in detecting bias within the model. We will explore mitigating bias from GPT and incorporating it into our task as part of future studies, as it is a completely new direction to our work. (Limitations section in Appendix J).
> >
> > > Q8. In general, a set of keywords is hardly sufficient to capture the semantic nuance that most memes carry. In the age of LLMs, one would expect an explainability approach to rely on some more sophisticated generative method to produce explanations that are readable.
> >
> > Yes, it is accurate to state that a set of keywords may not be sufficient to fully capture the semantic nuances of memes. However, our primary objective is to extract a set of keywords that are both 'interpretive' for model decision and 'plausible' with respect to the input meme. While LLMs generate more plausible natural language text that is readable, they often hallucinate, and it's challenging to verify whether the predicted text truly captures the inner workings of the model ('faithfulness'). The tradeoff between plausibility and faithfulness in natural language text from language models is well-discussed in Hase et al., 2020. In contrast, our method produces both plausible and faithful outputs.
> >
> > 1. **Are the extracted keywords faithful?** Yes. They are faithful to the model prediction. To validate this, we conducted an empirical experiment during testing: i) Initially, we assessed the model's confidence in the predicted class, and ii) Subsequently, we introduced knowledge tokens with the top 4 retrieved keywords from our framework. If these tokens truly interpret the model's inner workings, confidence in the initially predicted class should rise. Our findings demonstrate a positive numerical difference between the two measures for 97% of samples. Please refer to section 4.2.1 point VI in the updated paper for a detailed empirical discussion.
> >
> > 2. **Are the set of retrieved keywords plausible?** Yes, in short words. Our framework incorporates semantic matching through CLIP embedding as the third filtering step. This ensures that only keywords closely related to the input meme, leading to highly plausible outputs, are present in the retrieved keywords. Secondly, the GPT-2 prompt concludes with 'This meme is …,' deliberately chosen to guide subsequent tokens in accurately describing the input meme and its characteristics. Although the training strategy suggests that the most likely next tokens are 'normal' or 'hateful,' this does not prevent the retrieval of plausible explanatory keywords related to the meme. Additionally, in Appendix Table 4, we observe that keywords from our proposed method are more plausible in relation to the ground truth than those extracted from LlaVa-1.5-13B. For a comprehensive discussion, please refer to Appendix Section F in the updated paper. But it is also true that our model is limited in the sense that it cannot produce Natural Language explanations that are readable. We discuss these limitations in Appendix Section J.

---

> ### Comment · Reviewer_Diur · 2023-11-20
> **Response to author's revision**
>
> I have reviewed the author's responses and the associated revised version. While I appreciate the effort taken to improve the manuscript and take into account my comments, I am afraid that I cannot upgrade my score as I consider the performed updates to be insufficient. More specifically:
>
> The newly added introduction paragraph about the high level description of the inner workings is vague and does not provide the key methodological points as requested. The Appendix Figure 4 and section C provide more information and are in the correct direction but are still of low quality in terms of language as well as visual illustration. Further improvements are required.
>
> The literature review after the added papers is still in literally the same state as it was before the first review round. No description of the current state of the art and no analysis on what are the main methodological choices/directions in the literature. The authors just added a few more papers without explaining the methods nor highlighting how they address their shortcomings with this work.
>
> The use of language throughout the paper is still of low quality and needs a lot of further refinements.
>
> The methodology, although comprehensive, remains confusing and hard-to-read with poor mathematical representation of the considered components. The presentation of it requires effort towards refinement.
>
> The empirical experiment conducted in response to Q6 is not adequate. The concern is about the lack of explanation wrt the interaction between modalities. That is, associating certain image features/objects/concepts with parts of the overlay text that together construct hate speech is not addressed. In many cases the image and the text in isolation do not produce hateful outcome, and only when combined they do. So revealing this association and grounding it based on the interplay between the visual and textual signals is a major goal of the field. The fact that this is not addressed I think is a methodological fault of the proposed framework.
>
> Regarding OFA, the main concern (not addressed in the authors' answer) is that it very frequently provides an answer containing OCR on the meme's text rather an actual legit background image description. Is this encountered in the authors' experiments? Was it taken under examination?

---

> ### Author Response · Authors · 2023-11-21
> **Answer to followup questions**
>
> Thank you very much for the additional feedback. We very much appreciate the constructive criticism. We hereby respond to your follow-up questions as follows:
>
> > The newly added introduction paragraph about the high level description of the inner workings is vague and does not provide the key methodological points as requested. The Appendix Figure 4 and section C provide more information and are in the correct direction but are still of low quality in terms of language as well as visual illustration. Further improvements are required
>
> We have completely rephrased Section 3.1 (system design) in the main paper and streamined the detailed description in Appendix section C. Please check and kindly suggest if anything needs improvement.
>
> > The literature review after the added papers is still in literally the same state as it was before the first review round. No description of the current state of the art and no analysis on what are the main methodological choices/directions in the literature. The authors just added a few more papers without explaining the methods nor highlighting how they address their shortcomings with this work.
>
> Thanks for the feedback. Now we updated the related work section with more detailed explanation of the newly added papers.
>
> > The use of language throughout the paper is still of low quality and needs a lot of further refinements.
>
> In short time for author response, we tried our best to reformat and rewrite sections of the paper. We wholeheartedly take your suggestion and will keep improving the manuscript for final acceptance.
>
> > The methodology, although comprehensive, remains confusing and hard-to-read with poor mathematical representation of the considered components. The presentation of it requires effort towards refinement.
>
> Please check the updated manuscript, Section 3.1
>
> > The empirical experiment conducted in response to Q6 is not adequate. The concern is about the lack of explanation wrt the interaction between modalities. That is, associating certain image features/objects/concepts with parts of the overlay text that together construct hate speech is not addressed. In many cases the image and the text in isolation do not produce hateful outcome, and only when combined they do. So revealing this association and grounding it based on the interplay between the visual and textual signals is a major goal of the field. The fact that this is not addressed I think is a methodological fault of the proposed framework.
>
> Sorry for not understanding Q6 correctly. We added a section K in the Appendix which is named as 'Modality Importance'. We both quantitatively and qualitatively analyse both Multimodal as well as text and vision only models. Some takeaways are: i) Multimodal model performance is much better than either of text and visual only models. ii) The quality of representation is much richer for multimodal model. Whereas text and vision only models do perform good, sometimes the extracted keywords are either redundant or out of context. Please check Table 7 and Table 8 in the Appendix of the updated paper.
>
> > Regarding OFA, the main concern (not addressed in the authors' answer) is that it very frequently provides an answer containing OCR on the meme's text rather an actual legit background image description. Is this encountered in the authors' experiments? Was it taken under examination?
>
> Yes, we know and are aware of the fact that OFA generated captions are not good quality. While these captions are generally accurate, they may occasionally lack in-depth information. To address this limitation and boost the richness of the captions, we incorporate external knowledge tokens from both the Google Cloud Vision API and WikiData (as detailed in Appendix Section E). This external knowledge tokens compensate for the lack of quality on OFA generated caption.
>
> For instance, let's take a meme featuring Donald Trump from our training set (meme id: 90126). The OFA caption for this meme accurately describes the scene as "the picture of a man standing beside the American flag." However, it does not explicitly identify the man as Donald Trump. To bridge this informational gap, we integrate relevant external knowledge linked to this meme, such as "[KB] Donald Trump served as the president of the United States from 2017 to 2021." The OFA-generated caption and the external knowledge are then merged and presented as input to the GPT model. So the onput to GPT becomes "[CAPTION] the picture of a man standing beside the American flag. [KB] Donald Trump served as the president of the United States from 2017 to 2021.". This whole sentence is definitely better than the OFA generated caption alone. This is the reason we use knowledge text due to the very reason you also told that OFA generated captions are most of the time of low quality.
>
> We once again thank you for the review. Please feel free to ask questions for clarification.

---

> ### Comment · Reviewer_HAZ8 · 2023-11-22
> **Feedback on authors' response and final remarks [1/3]**
>
> **Summary of the feedback**
>
> I thank the authors for the detailed responses. Based on the responses and the corresponding questions posed, I am satisfied with the authors' responses to all the Suggestions made (Suggestions 1-11), and 11 Questions (Questions: Q1, Q3, Q4, Q5a, Q9, Q11a, Q11b, Q12, Q15, and Q17) posed while being partially satisfied with 3 Questions (Questions: 6, 7, and 8). I also reserve differences in the opinions expressed in the responses for all 7 Weaknesses highlighted and 5 Questions (Questions: Q2, Q10, Q13, Q14, and Q16) posed. Based on this, I am not going to update my rating. I feel the authors should incorporate all the suggestions and submit it later.
>
> ** Detailed comments **
>
> **W1, W6, W7:**	"It could be just a difference of opinion, but for me, this work is distinctly aligned more towards explainability (as it helps provide more context to decision making), and not so much as a case of interpretability as it isn't assisting in a direct/indirect way towards understanding the inner workings of the model towards making a decision. While what you are referring to as "interpretive nature" of the extracted keywords is actually "additional context" for the meme's content for me.
>
> The reason why it appears trivial is that there are strong, established reasons for realizing token sequence using strategies like beam-search, greedy search, sampling, etc., in the interest of narrowing down the output possible likely sequence space, and not because a larger sub-set of likely token sequence cannot be generated (along with additional pre-processing steps for better semantic alignment). Therefore, in principle, the proposed approach might not be as different from regular context generation via generative modeling, and yet not as effective as the latter, questioning the overall contribution value.
> While on the other hand, the proposed approach might not be an as well-contested solution to have while being modular in its design (with additional dependencies on the respective qualities of sub-modules), the majority part of which could be considered as additional post-processing steps to refine the model outputs, vs. highly effective and generalizable, end-to-end, multimodal LLM systems (subject to the quality of the LLM leveraged). It's not that the proposed solution is not yielding relevant outputs; its just the structural complexity and dependencies it has vs the simplicity along with the power that MLLMs bring, especially with highly efficient strategies like in-context learning, instruction fine-tuning, even zero-shot capabilities, etc., facilitated by the availability of foundation LLMs, just like the availability of pre-trained GPT2 and CLIP models.
>
> As part of your update on ""Comparison with MLLM"", the LAS for LLaVA is observed to be higher, but you also mention ""A higher LAS score obtained by our model"" in App F. A typo?"
>
> **W2 and W5, Point # 1:** The observations reflected from the empirical analysis conducted as part of the update introduced in Sec 4.2.1 are very much expected, as it involves re-conditioning the LM towards emulating the predictions, using the set of keywords, that were systematically curated as part of the overall keyword extraction pipeline in an earlier stage. The question is, how insightful is this observation?
>
> **W3 and W4:** Your additional updates towards generalizability all the more corroborate my apprehension of your proposed approach not being too generalizable. You had to clamp two prominent categories: 'very harmful' and 'somewhat harmful', into a single category in-order to evaluate your proposed approach, highlighting it's inherent constraint. Even considering the potential assertion that it is fundamentally designed to cater to a binary classification setup, it would still be interesting to investigate how effectively can the proposed approach solve 'very harmful' vs 'partially harmful', or an 'explicit hateful' vs 'implicit hateful' classification tasks, without yielding a coherent contextual evidence.
>
> Contd.....

---

> ### Comment · Reviewer_HAZ8 · 2023-11-22
> **Feedback on authors' response and final remarks [2/3]**
>
> **W5 and W6:**	*Point # 2:* Plausibility can be dynamic for MLLMs. The measure of 'plausability' in this case, is based upon the quality of keywords derived, which in turn is directly conditioned upon the 'type and quality of the prompt' you are probing the MLLM with in case of LLaVA. The fact that your prompt explicitly asked the MLLM to generate keywprds while towards the objective of ""summarizing"" its assimilation about the meme's context, significantly biases the output quality, which might not aptly match the level of abstraction that EXPLEME's outputs represent. This could have been easily addressed by an optimal (better) prompt engineering towards emulating the expected outputs. Moreover, the magnitude of the rapid advancements accompanying the new LLM (multimodal) will instantly side-step the limitations observed. The idea is not to force competition with the technological pace but to build something on top of the SOTA, to leverage its capacity, especially when it is inducing paradigm shifts.
>
> **W5 and W6:** *Point #3*: While I do acknowledge that your zero-shot OOD generalzability is decent, but since you state that your proposed approach is ""completely unsupervised"" by design, the initial manuscript solicited an acknowledgement of this fact, like stating the ""autoregressive language modeling"" objective for generating text to the least, hence completing the discussion about it. This requirement becomes all the more legit, when your approach involves trainable matrices and feed-forward neural networks in the early stages, which would necessitate the discussion of the overall training objectives, but your paper seems to have discussed little about such details."
>
>
> **Q2:** "Instead of merely explaining the meme's content," --> You might want to take another look at Sharma et  al., 2022. They do investigate implicit context conextualization, while factoring-in and modeling the categorical phenomena (hero/villain/victim), while proposing a MTL framework that is trained in an end-to-end manner, hence the question of imperfect positioning towards the possible gap.
>
> **Q6:**	For sub-part (a) -- The meme text could likely be leveraged (subject to empirical examination) towards identifying semantically relevant keywords via either (1) CLIP text-text, OR (2) transformer text-text interactions.
>
> **Q7:**	For sub-part (b) -- To rephrase the question in simple terms: Did you think of the generated knowledge tokens corresponding to the visual description (or the visual aspects) of the meme, towards conditioning (prompting) your LM as part of the final step of keyword retrieval? As this is the case in many nuanced meme scenarios, where visual consideration becomes dominant towards the final asnwer adjudication
>
> **Q8:**	For sub-part (b) -- By standard Alignment vs Optimization tradeoff, I am essentially referring to the standard techniques and the typical trade-offs they (vector alignment and loss curve optimization-based paradigms) would induce when considered jointly, and NOT something that goes by the exact term. Indeed your work does seem to be introducing the term at first glance, but the main question I posed, still remains, how is the proposed trade-off any different from what has already been explored pertaining to "Alignment" and "Optimization", when factored in jointly, for instance within the related contexts of autoregressive language modellng, neural sequence modelling, language modeling in general, etc.?
>
> **Q10:**	Your response justifies the observations made via qualitative analysis and theoretical argument, but not the ambiguity due to empirically observed (Table 1) similarity of observations in enabling vs disabling of e-ball constraint, which was the primary point raised in the question posed.
>
> **Q13:**	The question is not about whether you have done deduplication or not, instead it seeks to know your opinion on whether a special processing step like deduplication is applied on the output of CLIP-only (presuming it is capable of generating more than just 'jew'), and some other sampling strategy. I am presuming you don't really have different variants of your keyword extraction steps compared in Table 3 and you basically compare with several baseline outputs. So in other words, can customized filtering/sampling steps when applied systematically as per the model's requirement (especially considering CLIP, with its impressive zero-shot multimodal representation capability), yield reasonably good keywords?
>
> **Q14**	The question isn't implying examining GPT2 in isolation, instead, it seeks clarity on the possibility of having an end-to-end learnable solution at the interface of the CLIP multimodal output and GPT2 input and its implications rather than encoding of a binary signal-based, label mapping. Simply put: the implications of a jointly learnt framework vs. explicitly mapped and then embedded one.
>
> Contd...

---

> > ### Comment · Reviewer_HAZ8 · 2023-11-22
> > **Feedback on authors' response and final remarks [3/3]**
> >
> > **Q16:**	"The primary objective of the paper is not to provide explanations for the memes themselves..." --> And that is why the question doesn't talk about explanation at all; instead, it seeks the assessment of "keyword extraction quality" via the proposed generative language modeling setup (EXPLEME), when considering the biased distribution of various connotation within memes that exist for a fact (for example a lot of US political memes would portray Donald Trump in a negative light). How would the proposed approach deal with when exceptional variations of such cases are evaluated? Would the EXPLEME yield negative-connotation keywords for a meme about Mahatma Gandhi that criticises his ideals in a sarcastic manner within a distribution where most Mahatma Gandhi memes are positively narrated?

---

### Official Review · Reviewer_HAZ8 · 2023-10-26

**Soundness:** 1 poor
**Presentation:** 2 fair
**Contribution:** 1 poor
**Rating:** 3
**Confidence:** 4

**Summary:**

The paper discusses the challenge of detecting offensive content in memes, highlighting the shortcomings of binary classification and the need for reliable and unbiased classifiers. The authors propose a novel approach for extracting meaningful and interpretable tokens reflecting the meme’s content, enhancing model transparency and user trust. Their proposed approach constitutes four different stages designed towards the objective of extracting meaningful tokens. The stages include: (a) selecting the most relevant keywords, (b) filtering out data points outside a neighborhood of relevance, (c) ensuring semantic relevance within the candidate keywords, and (d) retaining output-preserving tokens from the candidates as the final set. Their method outperforms conventional interpretability baselines, contributing to meme content analysis and developing interpretable multimodal systems.

**Strengths:**

1. This paper is well-written and easy to understand. The proposed method is intuitive and built over multiple connected modules.

2. The approach of using auto-regressive language modeling loss, towards generating classification labels while learning to explicitly induce binary signals as part of the modeling approach is interesting. Although, it's a bit of a question itself as to the reliability of manually mapping these signals to the discrete labels, and then further using them for generative LM-based processing. Hence, it could add to the existing limitations of the approach.

**Weaknesses:**

1. A rudimentary approach to explainability and contextualization (being discussed from the lens of interpretability) within the context of memes.

2. Although the framework *EXPLEME* extracts implicit but relevant keywords w.r.t. the input meme, this at times might not be sufficient to assist in the process of interpreting the model decision, as it could be viewed as adding additional related keywords, yet without imparting a coherent context to the intended implicit content.

3. The approach may have some limitations in terms of generalizing for harmfulness categories within the wild (prior distributions are unknown), as the label signals depend on a ‘dix’ (category mapping dictionary). It would be better to have more insights into these aspects.

4. The generalizability of the proposed approach, even though an aspect that can surely be investigated, given the variety of the harmfulness and thematic domains and meme design types, is severely lacking in this work.

5. Interpretability is typically examined while understanding or probing the model decisions (predictions) and breaking down the understanding in terms of the steps or key attributes of the modeling approach (framework) that led to a particular output. This also involves looking at and considering the given input in an as-is manner.
While your work does explore the aspect of understanding the model output in a better way, it does pose the question of whether it is the interpretability or explainability that it primarily intends to address. Even Hase et al., 2020, as cited in the paper, trace the inspiration of LAS to interpretability but build the motivation based on the lack of work towards the ‘simulatability’ of NL explanations, which is a distinct notion from conventional interpretability.
From what I see it, generating semantically relevant textual cues by leveraging the inherent sampling strategy fundamental to generative LM might not necessarily facilitate model interpretability and could offer some form of explainability in some sense (maybe the term simulatability could be popularized with such paradigms), which has been recently reported in the literature to have made significant strides into, in terms of contextualizing the implicit content conveyed within memes [1] and Desai et al., 2021 and Sharma et al., 2022, as has also been cited in the current work’s literature, essentially questioning the research-gap established.

6. With noteworthy strides demonstrated by multimodal LLMs (MLLMs) like LLaVA [2], miniGPT4 [3], etc., apparently, there is no reference, citation and discussion, w.r.t positioning the proposed solution or comparison with it. As multimodal LLMs have demonstrated remarkable capacity towards not only describing the surface level details pertaining to visual-linguistic grounding but also subtle nuances (reasoned via the LLM attached) conveyed/implied within multimodal content, it becomes imperative to factor in their role while investigating solutions towards interpreting or contextualizing meme-like multimodal content.

7. The direct limitation of not examining or leveraging the latest capabilities of MLLMs is also observed in Table 3, third and fourth examples, which are classic examples of nuances involved in typical memes, which the proposed approach fails to resolve.

[1] MEMEX: Detecting Explanatory Evidence for Memes via Knowledge-Enriched Contextualization (Sharma et al., ACL 2023)
[2] https://llava-vl.github.io/
[3] https://minigpt-4.github.io/

**Questions:**

**Q1** *Introduction, third para, third sentence*: The text claims that the keywords linked to model predictions often don't semantically match the input meme. It's unclear if this assertion is based on empirical evidence or previous literature, and a citation for support is recommended. Additionally, while the importance of examining methods to contextualize textually expressed harm is highlighted, what is the take of authors on the consideration of implicit visual cues in memes and their integration into the proposed model framework?


**Q2** Is there anything else, in addition to contextualizing memetic phenomena under consideration, that your work is studying on a broader level? As taking a quick look at the outset of Desai et al., 2021 and Sharma et al., 2022, and their main objectives, they do seem to address multimodal contextualization for memes due to obscured meaning. So stating that “existing methods cannot fully explain the model behavior that is not directly related to the input but has some hidden meaning” might not help in positioning your attempts towards a possible gap that previous works seem to have touched upon.


**Q3** *Methodology*,
(a) first line: “The proposed systems combine” → Are there multiple “systems” that you’re proposing? Or your proposed approach/methodology/”system” combines a multimodal encoder and a language model. This is in line with the phrase “Our system follows a two-step strategy…” in the very next para. Please clarify and streamline.

(b) last line: “The incorporation of LM enables us to retrieve the set of explainable out-context keywords that are helpful in interpreting the system and its outcome.” → How do you position your goals and attempts against the recent developments within the field involving multimodal LLMs? How does your proposal compete/position/add on to what such multimodal LLMs can achieve?


**Q4** *Dimensions of the multimodal encoding:* How is it that you are working with the dimensions m X 1 and n X 1 for f_{t} and i_{t}, respectively, when one of the standard variants of pre-trained CLIP model: “openai/clip-vit-base-patch32”, renders a common dimension of 512, representing the joint multimodal representational space? Moreover, even if you are working with m and n as the first dimension sizes of these features, how come both U and V (trainable weight matrices) have x X ko as dimensions when your features represent different sizes? Kindly resolve the ambiguity.

**Q5** *Classifying via LM:*
(a) “li = argmax(FFN(Mt),dim = 1)”: What is the motivation behind employing an argmax, followed by explicit mapping of the signals to either Offensive vs normal, as against learning representations that could directly be used to condition the LM (GPT2) output? The question is additionally motivated by the general idea that performing argmax operation over FFN output is not typically recommended, as the model is usually trained w.r.t. a smooth loss function and doesn’t necessarily perform hard classification. Did you examine your intermediate signal outputs? Any empirical insights probing this aspect would shed some more light on the questionable reliability of this approach.

(b) “lab = SumPool(gl ◦ E[dix[l]])”: Does the small circle operator bw the gl and E terms represent element-wise multiplication? In either case, what is the motivation behind implementing interaction between gl output and E terms, when there’s no jointly learnt connection between the two? The effect intended to be captured via this operation isn’t super clear.

**Q6** *Sec 3.2, point about Semantic Relevance,*
(a) “the meme is encoded using the CLIP Vision encoder”: So the dot product computed bw CLIP text encoding of the keywords from the second step and CLIP visual embedding of the meme image doesn’t factor in the CLIP textual encoding of the meme text? Wouldn’t this lead to a relatively lossy multimodal embedding towards examining the semantic relevance?
(b) “First, we use the trained LM in inference mode”: Is it the LM trained as part of your experiments or the standardized (pre-trained/fine-tuned) one? This isn’t completely clear, and it could have implications on the type of output to expect.

**Q7** *Sec. 3.2*,
(a) “If the model predicts the same class as it predicted before”: As per my understanding, the previous prediction being referred to here in the LM’s primary output is to be considered as predicted (generated) label, thereby not suggesting any grounding w.r.t. the ground truth label. Can prediction flip for some scenarios suggest rectification of the previously incorrectly predicted (generated) label? Need more clarity here.

(b) “token does not have enough importance”: Do you observe any effect due to missing modality here? As when you reinforce your gen-LM’s output as knowledge bite in text-only form, there could be potential meme scenarios, where you end-up losing key information in terms of the intended subsequent LM’s conditioning.

**Q8** *Sec. 3.3, Alignment vs Optimization tradeoff,*
(a) “In practical applications, this serves as a filtering mechanism to retain tokens relevant to regions”: Is this an empirically verified finding or an established fact, in which case proper citations are a must?

(b) “We term this phenomenon the ‘Alignment vs. Optimization Trade Off Criteria’”: How does your proposed approach, in its scope, offer functionality or results in ways any different from the conventional implications posed by the standard “Alignment vs Optimization tradeoff”?

**Q9** *Sec 4.2.1, Comparison with the baselines:* I might have missed it, but what are the explanations being used for the explanation-based evaluation (F1 w/ exp) of the random/baselines? I presume the ones obtained (extracted) using the proposed approach are used for evaluating proposed (and variants). Now whatever may be the case for baselines, is it due to the ineffectiveness of the explanation derivation mechanism (hence the generated explanations) or of the interpretability model baselines themselves?


**Q10** As part of the results (Table 1), it is interesting to observe a consistent range of the diversity scores (inter/intra) and other metrics as well for e-ball enabled vs disabled scenarios, with top K enabled. How would the authors justify the relevance of the e-ball constraint as part of the proposed approach, with such consistent and barely distinct reproducibility with and without it?

**Q11** *Analysis, Does epsilon ball constraint…, page 7,*
(a) First line, “Without any ε constraint, we obtain a negative LAS score along with a low ‘comprehensiveness’ score.”: Table 1 suggests more on this, with higher ( and non-negative) LAS and comprehensiveness scores, without e-constraint cases (but with top-K enabled). Please resolve the ambiguity, and clarify the confusion.

(b) Last three lines: The optimal value of epsilon observed (0.01<0.05), suggests a smaller neighborhood, as an ideal scenario for better quality keyword extraction. On the contrary, the corresponding theoretical justification stated for it implies near orthogonality b/w e and delta_{m}f(m) components. Do the two have any direct connection in this scenario that I missed?


**Q12** *Fig. 2 and Analysis point # iv* “What is the similarity of the retrieved…”, page 8, third line: Firstly specifying that [-E,+E] represents E-neighbourhood, then suggesting Jaccard Similarity spikes at [-0.01, +0.01], renders reading Fig. 2 slightly difficult, as your y-axis is JS, and your x-axis has range [1,9], with x-axis label as E-neighbourhood. How to map the range of [-0.01, +0.01] on either of the axes with these configurations?

**Q13** Table 3, example 3 (91768), CLIP-only stage derives “jew, jew, jew, jew”, and ends up misclassifying a normal meme as an offensive one. Since your proposed approach constitutes several filtering stages, would considering some additional steps like post-processing by deduplicating the keywords generated (like in this case), would have facilitated the required diversity for it to be correctly classified?


**Q14** *Sec 4.3, last line:* Could a directly learnt input representation for GPT2 have given better results, as compared to an explicit transformation based categorical signal? Was the alternative explored as part of the investigation?


**Q15** Use Case: EXPLEME is designed to filter keywords that are directly linked with appeared entities in the meme. But if the same entity is used in another context, then will EXPLEME be able to perform well? For example, a meme contains an image of Donald Trump’s angry face towards the Mexico border, and simultaneously, the other side of the meme contains the image of Hitler’s smirking face with the text ‘Need suggestions !’. Although Hitler had no direct connection with Mexico and its border-related issues, when the smirk of Hitler is merged with the angry face of Donald Trump, it produces an implicit hate meme. Will EXPLEME be able to generate related keywords that connect both cases?

**Q16** Use Case: An entity such as a celebrity or a politician can be used in both positive and negative roles based on the context. If the entity has a biased association towards the negative connotation (example, Hitler) or a positive one (example, Mahatma Gandhi) but has been used in the opposite connotations, then would EXPLEME be able to suggest insightful interpretation/contextualization?

**Q17** The process of collecting external knowledge texts requires more elaboration. External knowledge snippets are fetched based on predefined tokens. Based on what heuristics are the tokens selected? How do the authors go about augmenting the knowledge with a few tokens that are related to the implicit facts?


**Suggestions/Clarifications**

**S1** *Abstract:*
(a) “However, binary classification of memes as offensive or not often falls short in practical applications.” → This can be generalized well beyond “offense detection”,  like for other hate speech related aspects/categories as well

(b) ‘“tokens” from a global vocabulary’ → It  might be prudent to characterize the scope of the vocabulary within the context of utilizing a particular LM in such scenarios, rather than suggesting it to represent a global vocabulary, which of course can have different technical implications.

**S2** Introduction, second para, last line, “...kind of inadvertent...“: “…kind of inadvertent biases…”?

**S3** *Related Work,*
(a) “when kiela2021hateful introduced a set of benchmarks”: Citation format issue.
(b) first para, last line: “This led to a number of research on de- tecting offensiveness in multimodal media, particularly in memes particularly in memes(Sharma et al., 2020; Kiela et al., 2021; Suryawanshi et al., 2020).” → There could be better citations supporting the argument of the follow-up developments, after Kiela et al., 2021.

**S4** The first formal mention about the task being addressed and the dataset being utilized is mentioned as part of the Sec 3, Classifying via LM and Sec 4.1, Experimental Setup. It is advisable to mention both at an earlier stage of the write-up to build the necessary backdrop, essential towards a complete understanding of the work and methodology.

**S5** You are addressing the task of offensive vs normal, upon which your entire contextualizing keyword extraction steps depends. It might not be recommended to conflate the concepts of “hate speech” and “offense”, when modeling one phenomena, while working with the dataset, build for the other. Offense is a broader term, and hate speech may involve some form of offense, but converse isn’t always true.

**S6** Sec 3.2, third last line, “linguistic tokens beyond those originally within”: Rephrase?

**S7** Sec 3.3, Alignment vs Optimization tradeoff, page 4, sixth last line, “gradient-based optimization steps of m”: or “gradient-based optimization steps WITH RESPECT TO m”?

**S8** You invest reasonable effort towards theoretically establishing and then analyzing alignment-related aspects of the trade-off while not so much on the optimisation front. Any relevant presumptions I missed in the write-up?

**S9** Sec 4.2.1, Comparison with the baselines, fifth line,
(a) “proposed approach by resorting to better obtaining scores”: Check the usage of the word resorting in this statement.

(b) “model in the next 10 rows,”: Table 1 shows 11 rows with similar configurations after the top section involving comparative baseline systems. The table can surely use better segregation and markers highlighting the proposed approach and its variants to avoid any confusion.

**S10** Fig. 3, Caption, various TopK and E values: Is the intention to probe various E values or simply compare E-constraint enable vs disable with the optimally set value of E? Kindly clarify.

**S11** An ablation probing the efficacy of the proposed model without the maximal and semantic relevance stages and only with epsilon-constraint with varying values of E would also be interesting.

---

> ### Author Response · Authors · 2023-11-19
> **Response to weaknesses for Reviewer HAZ8 (Part 1)**
>
> Thank you for the thoughtful review. We appreciate your insights and have taken steps to address the identified weaknesses and address your questions as outlined below.
>
> > A rudimentary approach to explainability and contextualization (being discussed from the lens of interpretability) within the context of memes.
>
> Ans: While the initial impression of our approach may seem rudimentary in terms of explainability and contextualization, a more in-depth analysis undertaken during the rebuttal phase reveals its nuanced effectiveness. This is demonstrated through the interpretive nature of the extracted keywords in informing model decisions, as detailed in the updated paper in Section 4.2.1 (specifically, point vi).
>
> Furthermore, to underscore the capabilities of our model, we conducted a comparative evaluation against an existing Multimodal Large Language Model (MLLM), namely Llava v1.5 13B. The results, presented in the updated paper in Section 4.2.2, showcase the comparable performance of our model to Llava 1.5 in both 'plausibility' and 'interpretability,' as gauged through the analysis of extracted keywords.
>
>
> > Although the framework EXPLEME extracts implicit but relevant keywords w.r.t. the input meme, this at times might not be sufficient to assist in the process of interpreting the model decision, as it could be viewed as adding additional related keywords, yet without imparting a coherent context to the intended implicit content.
>
> Ans: The concern raised regarding the potential insufficiency of EXPLEME's extracted keywords to aid in model interpretation is addressed through detailed empirical analysis, as outlined in the updated paper's Section 4.2.1, specifically in point VI.
>
> Simulatability, identified as a form of interpretability, is crucial in evaluating the faithfulness of an explanation to the model's behavior. Comparing two sets of outputs—more simulatable outputs (set A) versus less simulatable outputs (set B), measured by LAS—indicates that set A better supports the model's behavior. In our empirical evaluation, the keywords retrieved by EXPLEME (Set A) achieve significantly higher Leakage adjusted simulatability (LAS) scores than Set B, consisting of the existing input attribution-based methods. This underscores that our method's retrieved keywords serve as a strong indicator of model interpretability, particularly from a simulatability perspective.
>
> Highlighting the importance of faithfulness in interpretation, as emphasized by Hase et al. [1], we leverage simulatability-based metrics, such as LAS, to measure faithfulness. A higher LAS score, as observed in our method, signifies superior interpretability.
>
> To address the reviewer's concern regarding the interpretability of our method, we conducted an empirical experiment. At the testing time, i) Firstly, we measure the model’s confidence in the predicted class and then ii) Secondly, we introduce the knowledge tokens with the Top 4 retrieved keywords from our framework. The objective was to assess whether these tokens indeed contributed to the interpretation of the model's inner workings. Our findings reveal that, for 97.41% of test time samples, the observed difference between the confidence levels before and after the introduction of knowledge tokens was consistently a positive real number, which means that the extracted keywords increase model confidence on its predicted class, subsequently entailing that set of keywords is interpretive for model decision.
>
>
>
>
> > The approach may have some limitations in terms of generalizing for harmfulness categories within the wild (prior distributions are unknown), as the label signals depend on a ‘dix’ (category mapping dictionary). It would be better to have more insights into these aspects.
>
> Ans: To evaluate the generalizability of our approach, we utilized the HarMeme dataset, conducting zero-shot evaluation. Our model only classifies memes into either hateful or non-hateful categories, reflecting the binary nature of the training dataset. However, the keyword extraction process demonstrates generalizability across datasets (as can be seen through performance in FB meme dataset and HarMeme dataset) and diverse memes, as evidenced by examples and empirical evaluation.
>
> Both the classification performance (evaluated by clamping ‘somewhat harmful’ and ‘very harmful’ classes in one ‘harmful’ class) and the quality of extracted explainable keywords are well-enough to ensure that the proposed framework might be useful in the wild. More details can be found in Section 4.2.3 of the revised paper, and Appendix Section G.

---

> > ### Author Response · Authors · 2023-11-19
> > **Response to weaknesses for Reviewer HAZ8 (Part 2)**
> >
> > > The generalizability of the proposed approach, even though an aspect that can surely be investigated, given the variety of the harmfulness and thematic domains and meme design types, is severely lacking in this work.
> >
> > Ans: The generalizability of our proposed approach has been evaluated through the inclusion of an out-of-distribution (OOD) meme dataset, allowing for a zero-shot assessment of our model's performance. For detailed insights into this assessment, please refer to the updated paper, specifically in Section 4.2.3 and Appendix Section G, where additional clarifications and findings regarding the generalizability of our approach are provided.
> >
> > > Interpretability is typically examined while understanding or probing the model decisions (predictions) and breaking down the understanding in terms of the steps or key attributes of the modeling approach (framework) that led to a particular output. This also involves looking at and considering the given input in an as-is manner. While your work does explore the aspect of understanding the model output in a better way, it does pose the question of whether it is the interpretability or explainability that it primarily intends to address.......
> >
> > Ans: To address the inquiry about whether our work primarily aims at interpretability or explainability, we provide a comprehensive response in three key parts:
> > 1. **Does the extracted keyword help in interpreting model decision? Are they interpretive?** Yes. They are interpretive. We design an empirical experiment to verify it. At the testing time, i) Firstly, we measure the model’s confidence in the predicted class and then ii) Secondly, we introduce the knowledge tokens with the Top 4 retrieved keywords from our framework. In this analysis, we observed that when introducing knowledge tokens comprising the Top 4 retrieved keywords from our framework, the confidence in the previously predicted class increased. This effect was observed in 97.41% of the test time samples, with the difference between the latter and the former being a positive real number. A comprehensive and detailed empirical discussion of this analysis is provided in Section 4.2.1, point VI, of the updated paper.
> >
> > 2. **Whether the set of retrieved keywords really explains the meme?** Yes. Our framework incorporates semantic matching through CLIP embedding as the third filtering step, ensuring that only keywords closely related to the input meme (representing highly plausible outputs) are included in the retrieved keywords. Additionally, the GPT-2 prompt is intentionally structured to conclude with ‘This meme is …’, guiding the subsequent tokens to accurately describe the input meme and its properties. While the training strategy indicates that the next most plausible tokens are typically ‘normal’ or ‘hateful’, this does not preclude the retrieval of plausible explanatory keywords associated with the meme. This observation is further supported by the results presented in Appendix Table 4, where keywords obtained from our proposed method are notably more plausible than those extracted by LlaVa-1.5-13B.
> > For a comprehensive contextual discussion, please refer to Appendix Section F in the updated paper.
> >
> > 3. **What is the research gap established?** The research gap addressed by our method lies in its completely unsupervised nature, distinguishing it from existing research such as Sharma et al. (2022) and Desai et al. (2021), which rely on custom-built supervised datasets for training specific downstream systems. The unique contributions of our approach are summarized as follows:
> > a) Unsupervised Nature: Our method is entirely unsupervised, offering a distinct advantage in contrast to methodologies that depend on custom datasets and supervised training.
> > b) Explainability and Interpretability: The extracted keywords from our framework demonstrate both explainability and interpretability concerning the input meme, as outlined in points 1 and 2 above.
> > c) OOD Generalizability: Our method showcases out-of-distribution (OOD) generalizability, as discussed in detail in Section 4.2.2 of the updated paper.
> > d) Comparability to State-of-the-Art: Despite having significantly fewer parameters, our method is comparable to state-of-the-art models, as elaborated in Section 4.2.3 of the updated paper.
> >
> > While acknowledging these strengths, it's important to note that our method currently lacks support for natural language sentence output. However, we recognize this limitation and aim to address it in future work.

---

> > > ### Author Response · Authors · 2023-11-19
> > > **Response to weaknesses for Reviewer HAZ8 (Part 3)**
> > >
> > > > With noteworthy strides demonstrated by multimodal LLMs (MLLMs) like LLaVA [2], miniGPT4 [3], etc., apparently, there is no reference, citation and discussion, w.r.t positioning the proposed solution or comparison with it. As multimodal LLMs have demonstrated remarkable capacity towards not only describing the surface level details pertaining to visual-linguistic grounding but also subtle nuances (reasoned via the LLM attached) conveyed/implied within multimodal content, it becomes imperative to factor in their role while investigating solutions towards interpreting or contextualizing meme-like multimodal content.
> > > > The direct limitation of not examining or leveraging the latest capabilities of MLLMs is also observed in Table 3, third and fourth examples, which are classic examples of nuances involved in typical memes, which the proposed approach fails to resolve.
> > >
> > > Ans: We acknowledge the significance of multimodal Large Language Models (MLLMs) like LLaVA [2], miniGPT4 [3], and others in their remarkable capacity to comprehend and describe both surface-level details and subtle nuances within multimodal content. However, we would like to emphasize that our proposed method serves a distinct purpose (​​developing an unsupervised framework by augmenting a classifier via a language model to extract ‘plausible’ and ‘interpretable’ keywords that would simultaneously explain the input as well as interpret model decision), and direct comparison with MLLMs is not entirely equitable.
> > >
> > > There are key differentiators that warrant a nuanced consideration. Firstly, MLLMs, such as LLaVa-1.5-13B, entail manyfold more parameters than our model (LlaMa-1.5-13B is 26 times larger than our system) and are trained on vast amounts of data, whereas our framework operates in an unsupervised manner. Additionally, our method lacks the technical capabilities supporting natural language sentence outputs.
> > >
> > > The primary research gap addressed in this paper revolves around the development of an unsupervised framework, augmenting a classifier via a language model to extract 'plausible' and 'interpretable' keywords. These keywords serve the dual purpose of explaining the input and interpreting the model decision. In our comparative analysis, we focus on benchmarking our method against the state-of-the-art (LlaVa-1.5-13B) with significantly more parameters, as detailed in Section 4.2.3 of the updated paper. This comparison highlights that both methods achieve comparable performance when assessed through the lens of extracted keywords.

---

> > > > ### Author Response · Authors · 2023-11-19
> > > > **Response to questions for Reviewer HAZ8 (Part 1)**
> > > >
> > > > > Q1 Introduction, third para, third sentence: The text claims that the keywords linked to model predictions often don't semantically match the input meme. It's unclear if this assertion is based on empirical evidence or previous literature, and a citation for support is recommended. Additionally, while the importance of examining methods to contextualize textually expressed harm is highlighted, what is the take of authors on the consideration of implicit visual cues in memes and their integration into the proposed model framework?
> > > >
> > > > Ans: The statement is grounded in empirical evidence, supported by examples presented in Table 3, which demonstrate that existing input attribution methods fail to accurately capture the semantics of the meme.
> > > > Regarding implicit visual cues, our model addresses this aspect through a multi-step process. The interaction between visual and textual representations is managed using CLIP, and multimodal interaction is facilitated through the projection followed by an average-pooling module. This combined representation is then seamlessly integrated into the GPT-2 model. Additionally, textual descriptions of visual cues are incorporated through the inclusion of image captions and knowledge keywords, ensuring a comprehensive approach to capturing both textual and visual elements within memes.
> > > >
> > > >
> > > > > Q2 Is there anything else, in addition to contextualizing memetic phenomena under consideration, that your work is studying on a broader level? As taking a quick look at the outset of Desai et al., 2021 and Sharma et al., 2022, and their main objectives, they do seem to address multimodal contextualization for memes due to obscured meaning. So stating that “existing methods cannot fully explain the model behavior that is not directly related to the input but has some hidden meaning” might not help in positioning your attempts towards a possible gap that previous works seem to have touched upon.
> > > >
> > > > Ans: Both papers focus on providing a natural language description of why a meme is perceived as hateful. However, it's important to note that our approach diverges in its objective. Instead of merely explaining the meme's content, our method is designed to justify the model's decision by highlighting main keywords contributing to the chosen decision.
> > > > Our method does not need external datasets and it's completely unsupervised. Basically it is proposed as an augmentation over existing classifiers via a language model.
> > > > We empirically show that our proposed method is both interpretable and explainable (see section 4.2.1 point 6 and 4.2.2 of the updated paper). We compare our proposed method wrt an existing large VLM (LlaVA 1.5 13B) and both seem to work more or less similar when meme interpretability and explainability are considered (see sec 4.2.2 of the updated paper).
> > > >
> > > > > Q3 Methodology, (a) first line: “The proposed systems combine” → Are there multiple “systems” that you’re proposing? Or your proposed approach/methodology/”system” combines a multimodal encoder and a language model. This is in line with the phrase “Our system follows a two-step strategy…” in the very next para. Please clarify and streamline.
> > > > (b) last line: “The incorporation of LM enables us to retrieve the set of explainable out-context keywords that are helpful in interpreting the system and its outcome.” → How do you position your goals and attempts against the recent developments within the field involving multimodal LLMs? How does your proposal compete/position/add on to what such multimodal LLMs can achieve?
> > > >
> > > >
> > > > Ans: a) The strategy involves a conceptual division into two steps, but the implemented model is an end-to-end system. While the process can be visualized as having two steps, all predictions and functionalities stem from a unified system, where a classifier is augmented by a Language Model (LLM). In essence, the model seamlessly performs both classification and explainable token extraction as integrated tasks.
> > > >
> > > > b) Our method is systematically compared with an existing Vision-Language Model (VLM), specifically LlaVa 1.5 13B. Comprehensive results of this comparison have been incorporated in Section 4.2.2, providing a detailed examination of the findings in Appendix Section F.
> > > >
> > > >
> > > > > Q4 Dimensions of the multimodal encoding: How is it that you are working with the dimensions m X 1 and n X 1 for f_{t} and i_{t}, respectively, when one of the standard variants of pre-trained CLIP model: “openai/clip-vit-base-patch32”, renders a common dimension of 512, representing the joint multimodal representational space? Moreover, even if you are working with m and n as the first dimension sizes of these features, how come both U and V (trainable weight matrices) have x X ko as dimensions when your features represent different sizes? Kindly resolve the ambiguity.
> > > >
> > > > Ans: We appreciate your keen observation and want to acknowledge an oversight in our notation. Actually, n=m=512. 'n' and 'm' were chosen to distinguish between them.

---

> > > > > ### Author Response · Authors · 2023-11-19
> > > > > **Response to questions for Reviewer HAZ8 (Part 2)**
> > > > >
> > > > > > Q5 Classifying via LM: (a) “li = argmax(FFN(Mt),dim = 1)”: What is the motivation behind employing an argmax, followed by explicit mapping of the signals to either Offensive vs normal, as against learning representations that could directly be used to condition the LM (GPT2) output? The question is additionally motivated by the general idea that performing argmax operation over FFN output is not typically recommended, as the model is usually trained w.r.t. a smooth loss function and doesn’t necessarily perform hard classification. Did you examine your intermediate signal outputs? Any empirical insights probing this aspect would shed some more light on the questionable reliability of this approach.
> > > > >
> > > > >
> > > > > Ans: In our training approach, the Feedforward Neural Network (FFN) undergoes simultaneous training with GPT-2. Two essential components are supplied during this training process:
> > > > > Firstly, the argmaxed value, extracted from the FFN's output, is verbalized and then input into GPT-2 for further processing. Additionally, a continuous representation denoted as M_t is provided. This continuous representation captures nuanced data characteristics and plays a vital role in the overall learning process.
> > > > > To ensure effective gradient flow and optimize learning, the input embedding corresponding to the argmaxed label is subjected to the gumbel softmax operation. This step is crucial for retaining gradient information. By providing both argmaxed values and continuous representations, our training methodology aims to create a comprehensive and effective learning process.
> > > > >
> > > > >
> > > > > > Q5 (b) “lab = SumPool(gl ◦ E[dix[l]])”: Does the small circle operator bw the gl and E terms represent element-wise multiplication? In either case, what is the motivation behind implementing interaction between gl output and E terms, when there’s no jointly learnt connection between the two? The effect intended to be captured via this operation isn’t super clear.
> > > > >
> > > > > Ans: In our notation, 'gl' represents the gumbel logit, serving as a one-hot representation of the label 'l.' Notably, this representation is equipped with gradient information. Additionally, 'E[dix[i]]' is a stylized expression denoting the token embedding of the output class obtained from the Feedforward Neural Network (FFN). This output class can take on values such as 'normal' or 'offensive.' The updated Figure 4 in Appendix illustrates our idea. Please note the elaborate caption associated with Figure 4 and the description in Appendix Section C.
> > > > >
> > > > > > Q6 Sec 3.2, point about Semantic Relevance, (a) “the meme is encoded using the CLIP Vision encoder”: So the dot product computed bw CLIP text encoding of the keywords from the second step and CLIP visual embedding of the meme image doesn’t factor in the CLIP textual encoding of the meme text? Wouldn’t this lead to a relatively lossy multimodal embedding towards examining the semantic relevance? (b) “First, we use the trained LM in inference mode”: Is it the LM trained as part of your experiments or the standardized (pre-trained/fine-tuned) one? This isn’t completely clear, and it could have implications on the type of output to expect.
> > > > >
> > > > > Ans: a) Indeed, your observation aligns with our findings. While experimenting, we attempted using the concatenation of meme visual and textual embeddings to calculate the dot product. However, the downstream retrieved keywords exhibited poor quality. In contrast, relying solely on the visual embedding yielded much richer and more meaningful retrieved keywords. This outcome may be attributed to the contrastive learning approach employed in CLIP, where similar image-text pairs are effectively aligned.
> > > > >
> > > > > b) The Language Model (LM) is trained in an end-to-end fashion. Subsequently, the trained LM is employed in inference mode to generate keywords that can aptly describe the model's decisions. This two-phase process allows for effective training and utilization of the language model in extracting relevant and informative keywords during the inference stage.
> > > > >
> > > > > > Q7 Sec. 3.2, (a) “If the model predicts the same class as it predicted before”: As per my understanding, the previous prediction being referred to here in the LM’s primary output is to be considered as predicted (generated) label, thereby not suggesting any grounding w.r.t. the ground truth label...
> > > > >
> > > > > Ans: a) your emphasis on faithful and interpretable explanations is well-noted. It's prudent to disregard instances where the prediction changes from the initial guess during the extraction of keywords. This ensures that the extracted keywords remain consistent with the model's original prediction. The term "flipped prediction" accurately describes the situation where the extracted keywords fail to match the model's previous prediction. By rejecting prediction-flipping keywords, you maintain the interpretability of the explanations.
> > > > >
> > > > > b) Sorry, we could not understand. We would really appreciate it if you could clarify the question.

---

> > > > > > ### Author Response · Authors · 2023-11-19
> > > > > > **Response to questions for Reviewer HAZ8 (Part 3)**
> > > > > >
> > > > > > > Q8 Sec. 3.3, Alignment vs Optimization tradeoff, (a) “In practical applications, this serves as a filtering mechanism to retain tokens relevant to regions”: Is this an empirically verified finding or an established fact, in which case proper citations are a must? (b) “We term this phenomenon the ‘Alignment vs. Optimization Trade Off Criteria’”: How does your proposed approach, in its scope, offer functionality or results in ways any different from the conventional implications posed by the standard “Alignment vs Optimization tradeoff”?
> > > > > >
> > > > > > Ans: a) Empirical verification is presented in Figure 2, and the theoretical guarantee is elucidated in the proof outlined in Section 3.3, Theorem.
> > > > > > b) To the best of our knowledge, the term (“Alignment vs Optimization tradeoff”) introduced is considered novel. To the best of our understanding, there appears to be no recognized or standard "Alignment vs Optimization tradeoff" in existing literature. Your consideration and insights on this matter would be greatly appreciated.
> > > > > >
> > > > > >
> > > > > > > Q9 Sec 4.2.1, Comparison with the baselines: I might have missed it, but what are the explanations being used for the explanation-based evaluation (F1 w/ exp) of the random/baselines? I presume the ones obtained (extracted) using the proposed approach are used for evaluating proposed (and variants). Now whatever may be the case for baselines, is it due to the ineffectiveness of the explanation derivation mechanism (hence the generated explanations) or of the interpretability model baselines themselves?
> > > > > >
> > > > > > Ans: For the random baseline, we employ a randomly chosen set of four keywords. In the case of other baselines, we select four keywords from the input, considering that baseline methods rely on input attribution (e.g. Integrated Gradient). Additionally, the top four tokens are retrieved based on their input attribution scores. This choice is made not due to the ineffectiveness of explanation derivation mechanisms but as a deliberate selection for comparative analysis.
> > > > > >
> > > > > > > Q10 As part of the results (Table 1), it is interesting to observe a consistent range of the diversity scores (inter/intra) and other metrics as well for e-ball enabled vs disabled scenarios, with top K enabled. How would the authors justify the relevance of the e-ball constraint as part of the proposed approach, with such consistent and barely distinct reproducibility with and without it?
> > > > > >
> > > > > > Ans: The ϵ-ball constraint is a theoretically grounded mechanism that has demonstrated empirical success, especially when the topK constraint is enabled. Importantly, enabling the ϵ-ball constraint not only does not compromise the quality of the extracted keywords but serves as an additional filtering mechanism, contributing to the production of more faithful explanations.
> > > > > >
> > > > > > > Q11 Analysis, Does epsilon ball constraint…, page 7, (a) First line, “Without any ε constraint, we obtain a negative LAS score along with a low ‘comprehensiveness’ score.”: Table 1 suggests more on this, with higher ( and non-negative) LAS and comprehensiveness scores, without e-constraint cases (but with top-K enabled). Please resolve the ambiguity, and clarify the confusion. (b) Last three lines: The optimal value of epsilon observed (0.01<0.05), suggests a smaller neighborhood, as an ideal scenario for better quality keyword extraction. On the contrary, the corresponding theoretical justification stated for it implies near orthogonality b/w e and delta_{m}f(m) components. Do the two have any direct connection in this scenario that I missed?
> > > > > >
> > > > > >
> > > > > > Ans: a) TopK+e-ball > Top-K > e-ball as per the performance of the models. Also note that for each case, 3rd and fourth filtering stages were enabled.
> > > > > >
> > > > > > b) Near orthogonality b/w e and delta_{m}f(m) components signifies that the dot product between them (epsilon) will be more or less equal to zero. They are not contrary to each other.
> > > > > >
> > > > > >
> > > > > > > Q12 Fig. 2 and Analysis point # iv “What is the similarity of the retrieved…”, page 8, third line: Firstly specifying that [-E,+E] represents E-neighbourhood, then suggesting Jaccard Similarity spikes at [-0.01, +0.01], renders reading Fig. 2 slightly difficult, as your y-axis is JS, and your x-axis has range [1,9], with x-axis label as E-neighbourhood. How to map the range of [-0.01, +0.01] on either of the axes with these configurations?
> > > > > >
> > > > > > Ans: Each discrete point in the x-axis illustrates specific E-neighborhoods like [(-0.09,-0.07), (-0.07, -0.05),... (0.05,0.07)]. The mid-point in the x-axis refers to the E-neighborhood of [-0.01,+0.01]. It is also observed that around the midpoint in the x-axis there is a spike, essentially showing that jaccard similarity spikes at  around [-0.01,0.01].

---

> ### Author Response · Authors · 2023-11-19
> **Response to questions for Reviewer HAZ8 (Part 4)**
>
> > Q13 Table 3, example 3 (91768), CLIP-only stage derives “jew, jew, jew, jew”, and ends up misclassifying a normal meme as an offensive one. Since your proposed approach constitutes several filtering stages, would considering some additional steps like post-processing by deduplicating the keywords generated (like in this case), would have facilitated the required diversity for it to be correctly classified?
>
> Ans: We do not employ any deduplication filtering stage. Whatever is generated from the third stage is kept as it is both when i) first and second stages are enabled, ii) first and second stages are disabled. This entails a fare comparison of the ablated components to our proposed framework.
>
>
> > Q14 Sec 4.3, last line: Could a directly learnt input representation for GPT2 have given better results, as compared to an explicit transformation based categorical signal? Was the alternative explored as part of the investigation?
>
> Ans: We explored in this direction and the final classification results obtained from GPT2 are near random. The highest accuracy obtained using this standalone GPT2 training only gives 53% F1 score in the test dataset, whereas by employing our method, we obtain 79% test accuracy as can be seen from Table 1.
>
> > Q15 Use Case: EXPLEME is designed to filter keywords that are directly linked with appeared entities in the meme. But if the same entity is used in another context, then will EXPLEME be able to perform well? For example, a meme contains an image of Donald Trump’s angry face towards the Mexico border, and simultaneously, the other side of the meme contains the image of Hitler’s smirking face with the text ‘Need suggestions !’. Although Hitler had no direct connection with Mexico and its border-related issues, when the smirk of Hitler is merged with the angry face of Donald Trump, it produces an implicit hate meme. Will EXPLEME be able to generate related keywords that connect both cases?
>
> Ans: It is supposed to generate keywords like “genocide” and “invasion”, which relates both the cases and lead the model for the decision of the hateful classification. Upon careful inspection we did not found a meme like that in the internet, so we create a meme manually and upon giving it to our trained model, it is extracting these keywords: “refugee”, “mexico”, “murder” and “war”, which implicitly captures the nuances of the meme as well as interpretive of the model decision (which is offensive class)
>
>
> > Q16 Use Case: An entity such as a celebrity or a politician can be used in both positive and negative roles based on the context. If the entity has a biased association towards the negative connotation (example, Hitler) or a positive one (example, Mahatma Gandhi) but has been used in the opposite connotations, then would EXPLEME be able to suggest insightful interpretation/contextualization?
>
> Ans: The primary objective of the paper is not to provide explanations for the memes themselves. Instead, it focuses on elucidating the model's decision-making process. For instance, if the model classifies a meme as hateful and includes the term "Gandhi," it aims to reveal that the implicit decision of the model may be influenced by the presence of the entity "Gandhi" and how it contributes to the model's decision.
>
>
> > Q17 The process of collecting external knowledge texts requires more elaboration. External knowledge snippets are fetched based on predefined tokens. Based on what heuristics are the tokens selected? How do the authors go about augmenting the knowledge with a few tokens that are related to the implicit facts?
>
>
> Ans: Tokens are selected using google Cloud Vision API. It is elaborated on Appendix Section E. Firstly, we provide the meme to the cloud vision API to extract the relevant tokens. The WikiData is then used to extract contextual knowledge snippets for each token. The token and contextual knowledge snippet are then added which forms a knowledge text. For example,
> Knowledge tokens = “jew”
> Knowledge snippet = “jew are a member of the people and cultural community whose traditional religion is Judaism”
>
> Knowledge text = “[KB] jew are a member of the people and cultural community whose traditional religion is Judaism”

---

> ### Author Response · Authors · 2023-11-19
> **On Suggestions from reviewer HAZ8 and References**
>
> **Suggestions/Clarifications:**
>
> S1. a) Changed “However, binary classification of memes as offensive or not often falls short in practical applications.” -> “However, classification of memes as offensive or other categories often falls short in practical applications.” S2.b) Changed “tokens” from a global vocabulary’ -> “tokens” from a predefined vocabulary space’
>
>
>
> S2: Thanks. checked and added “biases”
>
> S3: a) checked.
>
> S4: added at the last paragraph of the introduction.
>
> S6: changed.
>
> S7: changed.
>
> S8: Please refer to 2nd point in section 3.3.
>
> S9: a) “resorting to” -> observing. b) Thanks for pointing out. We are now using two \hline for better segregation.
>
> S10: It is with the optimally set value of E. We added ‘fixed’ in the Figure caption to point out the mistake.
>
> S11: Without the third stage enabled, the quality of the retrieved keywords for every case is worse. We have written our outcome in Section 4.2.1 point VII.
>
>
> **References**
>
> [1] Peter Hase, Shiyue Zhang, Harry Xie, and Mohit Bansal. 2020. Leakage-Adjusted Simulatability: Can Models Generate Non-Trivial Explanations of Their Behavior in Natural Language?. In Findings of the Association for Computational Linguistics: EMNLP 2020, pages 4351–4367, Online. Association for Computational Linguistics.
>
> [2] Liu, H., Li, C., Wu, Q., & Lee, Y.J. (2023). Visual Instruction Tuning. ArXiv, abs/2304.08485.
>
> [3] Zhu, D., Chen, J., Shen, X., Li, X., & Elhoseiny, M. (2023). MiniGPT-4: Enhancing Vision-Language Understanding with Advanced Large Language Models. ArXiv, abs/2304.10592.
>
> [4] Shivam Sharma, Siddhant Agarwal, Tharun Suresh, Preslav Nakov, Md. Shad Akhtar, and Tanmoy Chakraborty. 2023. What do you MEME? generating explanations for visual semantic role labelling in memes. In Proceedings of the Thirty-Seventh AAAI Conference on Artificial Intelligence and Thirty-Fifth Conference on Innovative Applications of Artificial Intelligence and Thirteenth Symposium on Educational Advances in Artificial Intelligence (AAAI'23/IAAI'23/EAAI'23), Vol. 37. AAAI Press, Article 1097, 9763–9771. https://doi.org/10.1609/aaai.v37i8.26166
>
> [5] Desai, P., Chakraborty, T., & Akhtar, M.S. (2021). Nice perfume. How long did you marinate in it? Multimodal Sarcasm Explanation. ArXiv, abs/2112.04873.

---

> ### Author Response · Authors · 2023-11-21
> **Gentle Reminder to review Rebuttal**
>
> Dear reviewer,
> Thanks again for the detailed feedback. We have attempted to resolve your concerns through the rebuttal. We request you to kindly check and update your score accordingly. Please ask us any questions for clarifications. We would be very happy to answer.
>
> Thanking you.

---

### Official Review · Reviewer_BdWE · 2023-11-01

**Soundness:** 3 good
**Presentation:** 3 good
**Contribution:** 3 good
**Rating:** 8
**Confidence:** 4

**Summary:**

In this paper, the authors present a novel approach to extracting meaningful tokens from a global vocabulary for effective model interpretability for memes. They demonstrate the effectiveness of their approach on the Facebook Hateful Meme Dataset.

**Strengths:**

The paper is very well written. The proposed approach seems task-agnostic and would be relevant for model interpretability across meme understanding tasks such as harmfulness detection, offensiveness detection, etc. The authors perform an extensive evaluation to demonstrate the relevance of their approach.

**Weaknesses:**

It would be interesting to see the variation in the experiments in terms of the language model used (GPT2 currently), dataset for evaluation, tasks, etc. The proposed approach can be demonstrated to work across tasks, datasets, and language models (as claimed in the introduction/contribution).

**Questions:**

In the human evaluation, were the evaluators aware that they were evaluating a particular method (Ours, ϵ-ball, CLIP, Integrated Gradient)?

Also, I think that a section on the limitations of the proposed approach can be added (would be relevant for future works).

---

> ### Author Response · Authors · 2023-11-19
> **Response to Reviewer BdWE**
>
> Thank you very much for the positive feedback.
>
> **Response to Weakness**
>
> > It would be interesting to see the variation in the experiments in terms of the language model used (GPT2 currently), dataset for evaluation, tasks, etc. The proposed approach can be demonstrated to work across tasks, datasets, and language models (as claimed in the introduction/contribution).
>
> Ans: Thank you for your insightful suggestion. As you rightly noted, our proposed framework is designed to be versatile across various tasks, datasets, and language models, as highlighted in the introduction and contribution sections. In response to your recommendation, we conducted additional evaluations using the Harmeme dataset, employing a zero-shot approach. The qualitative analysis in Section 4.2.3 and Appendix Section G showcases the effectiveness of our method in extracting explainable keywords that capture both meme semantics and model interpretability.
>
> Furthermore, we acknowledge your interest in exploring other language models, and we plan to incorporate additional large language models (LLMs) in the camera-ready version of the paper during the rebuttal phase. It's worth noting that our initial choice of GPT-2 was influenced by its relatively low parameter count (355M), contributing to faster inference speeds. We appreciate your valuable input and strive to address these considerations in the final version of our work.
>
> **Response to Questions**
>
> > In the human evaluation, were the evaluators aware that they were evaluating a particular method (Ours, ϵ-ball, CLIP, Integrated Gradient)?
>
> We appreciate your concern about the evaluation process. The evaluators in the human evaluation were intentionally kept unaware of the specific method associated with the presented memes and explanations. To ensure an unbiased assessment, a script was employed to output memes and their corresponding explanations without revealing the corresponding method used for generating the explanations.
>
>
>
> > Also, I think that a section on the limitations of the proposed approach can be added (would be relevant for future works).
>
> Thank you for your suggestion. We have taken it into consideration and included a dedicated section on the limitations of our proposed approach. This valuable addition can be found in Appendix section J, providing a comprehensive overview of the potential constraints that should be acknowledged in the context of our work and serving as valuable guidance for future research initiatives.

---

### Official Review · Reviewer_VNYw · 2023-11-07

**Soundness:** 3 good
**Presentation:** 3 good
**Contribution:** 3 good
**Rating:** 8
**Confidence:** 2

**Summary:**

The authors present a method to automatically understand the content of a meme, which is a difficult task, because the text alone can be misinterpreted if the image is not taken into account (as the authors demonstrate with an example). Furthermore, the authors derive the mathematics for merging the text and image understanding and give a proof of their theorem. Finally, various experiments are conducted and analysed.

**Strengths:**

The paper presents maths and prove their theorem. Unfortunately, I wasn't able to fully check the math, and hence, also not able to fully verify and understand the results.

**Weaknesses:**

The human classification experiments are conducted by the authors of the paper. This should not be the case.

**Questions:**

I don't have any question.

---

> ### Author Response · Authors · 2023-11-19
> **Response to the Reviewer VNYw**
>
> Thank you very much for the feedback. Regarding the following weakness:
>
> > The human classification experiments are conducted by the authors of the paper. This should not be the case.
>
> Ans: Thanks for the insightful observation. We engaged two additional annotators from our research lab to reassess the 100 examples. The updated findings have been incorporated into Table 2 of the paper. Notably, the human evaluation now involves a panel of five individuals, and we observe a kappa score exceeding 0.7 across all setups, signifying a satisfactory level of inter-rater agreement. The human evaluators were intentionally kept unaware of the model associated with each sample to maintain unbiased feedback.

---

> > ### Comment · Reviewer_VNYw · 2023-11-20
> >
> > Dear authors, I appreciate the effort of including additional annotators. Unfortunately, there is still the potential of strong bias in the data.

---

### Author Response · Authors · 2023-11-19
**Common Response to Reviewers**

Dear Reviewers,

Thank you sincerely for your valuable and insightful suggestions. Your input has proven immensely beneficial in enhancing the quality and readability of our paper. We have diligently incorporated the recommended changes in the paper, and you will find the updated sections marked in blue. Additionally, we have repositioned the proofs to Appendix A and B for better organization, and we have made adjustments to Image 4 to enhance clarity. A high level working detail of our proposed model is discussed in detail in Appendix Section C.

Your constructive feedback has been invaluable, and we are grateful for the opportunity to refine our work based on your recommendations.